# Intra-group differences in skin tone influence evaluative and perceptual face processing

**Micah Amd** [ID] *

University of the South Pacific, Port Vila, Vanuatu

* micah.amd@proton.me

## Abstract

In an exploration of colorist biases across native Melanesian participants, we employed a multi-method approach across three studies to examine evaluative and perceptual processing of 'lighter' and 'darker' non-Melanesian facial targets controlled for attractiveness, sex, and ethnicity. In Study 1, 305 participants evaluated facial attractiveness using surveys. In Study 2, 153 participants alternately mapped lighter and darker faces with positive and neutral attributes across brief Implicit Association Tests. In Study 3, 61 participants underwent a manual sorting task followed by a 'breaking' continuous flash suppression (b-CFS) paradigm to probe 'non-conscious' perceptual biases. Across evaluative measures, male and female respondents consistently preferred lighter-skinned, highly attractive male faces. During b-CFS, lighter and attractive *opposite-sex* faces entered awareness ('broke suppression') faster than their darker counterparts. We speculate that skin tone may operate as a perceptually salient cue in the presence of facial configurations signaling high reproductive potential.

## Introduction

The social valuation of lighter-skinned individuals within racial groups, known as 'colorism,' is a pervasive and cross-cultural phenomenon reported across multiple Asian communities [1,2]. Colorist attitudes differ from racial attitudes in that the former occurs within racial groups, while the latter occurs between them [3]. Colorist preferences may manifest early in development, even in ethnically homogeneous, non-White communities [4,5]. According to [2], colorist preferences may be directed towards those with lighter skin tones in general ('classist' colorism) or towards lighter-toned persons who concurrently exhibit Eurocentric features ('colonial' colorism). In wider society, colorist preferences have been associated with reduced educational, career, and relationship prospects for individuals with darker skin tones, leading to dissatisfaction with one's natural tone and bolstering social acceptance of harmful skin-lightening products [6–8].

Research into colorist preferences within psychological science has been comparatively sparse compared to research on racial attitudes [2], a gap that the present research aimed to address. The current investigation explored whether colorist biases could be inferred along evaluative and perceptual levels of processing across samples of native Melanesians, who

**Data Availability Statement:** The data underlying the results presented in the study are available in an OSF file https://osf.io/fsbh3/.

**Funding:** MA received an internal research grant from the University of the South Pacific to conduct the current research. The funder had no role in

study design, data collection and analysis, decision to publish, or preparation of the manuscript.

**Competing interests:** The authors have declared that no competing interests exist.

constitute a non-*WEIRD* (Western Educated, Industrialized, Rich, and Democratic) and under-investigated demographic in a society that promotes and accepts colorism [9]. No previous study has empirically explored colorist attitudes across this demographic, let alone along multiple processing levels, highlighting the present work's importance.

We report three studies that explored how non-Melanesian faces were processed based on relative differences across the latter's skin tone. The decision to employ *non*-Melanesian faces was motivated by two factors: First, it is well-established that faces representing one's racial group are processed more effectively and evaluated more favorably than faces from other racial groups due to being encountered more frequently during development [10–12]. Second, there are currently no validated databases of Melanesian faces that have been controlled for attractiveness, emotionality, and/or other features known to influence face processing [13]. Both concerns were addressed by using non-Melanesian faces from a validated database [14] categorized along ethnicity (Asian, Black, Latin, White), sex (male, female), and attractiveness (low, high). Faces were classified as lighter or darker *within* each target category to ensure that any observed variance in outcome parameters was primarily attributable to skin tone over structural/morphometric differences [15].

We explored whether skin tone influenced facial processing under 'unconstrained' or 'constrained' responding conditions [16,17]. By 'unconstrained,' we imply response measures that allowed participants to freely deliberate before responding. This included participants in Study 1, who rated the attractiveness of faces using surveys, and participants in Study 3, who had to sort faces based on their subjective preference manually. Sorting tests can reliably distinguish between ordinally valenced stimulus categories like closed-ended surveys [17,18]. By 'ordinal,' we imply relatively ranked categories that are subjectively perceived as being more or less positively valenced relative to each other. Critically, both surveys and sorting allowed unlimited time and the option to revise initial responses. Alternatively, 'constrained' measures restrict top-down deliberation by, for example, requiring speeded responses between limited response options, as was the case for participants who currently underwent Brief Implicit Association Tests (BIATs) in Study 2.

Independent of response constraints on deliberation opportunity, both unconstrained surveys and constrained BIATs may ultimately capture *evaluative* responses, which describe propositions specifying relations between identified terms [19]. The identification of terms and the specification of any relationship between them presuppose conscious awareness [20,21]. In that case, *any* proposition(s) currently salient in conscious awareness, even if unrelated to colorist biases, can influence expressed evaluations, even when the latter are constructed under constrained responding conditions [22,23]. Because similar concerns can be levied against 'any' evaluative measure when considering evaluations are *de facto* conscious and mutable to potentially spurious propositions [24], we deployed a 'breaking' continuous flash suppression (*b*-CFS) protocol in our final study to assess whether lighter and darker faces were differentially processed before entering conscious processing [25].

## Breaking continuous flash suppression

During *b*-CFS, a target stimulus, such as a face, is presented to the non-dominant eye. The target is rendered temporarily invisible ('suppressed') from conscious awareness by simultaneously presenting rapidly changing ('flashing') visual patterns to the other eye [25]. The critical parameter is the time taken for suppressed stimuli to 'break through' and reach conscious awareness, as captured through (for example) target localization responses [26]. Faces that 'break suppression' faster may be comparatively privileged by perceptual, potentially nonconscious processes relative to faces that take longer to break suppression [25–28]. Deploying

*b*-CFS allowed us to explore whether biases towards face categories could be detected before the latter had entered awareness and rendered susceptible to conscious deliberate processes [16].

Our *b*-CFS protocol was adapted from previous designs for consistency [29–31]. This involved presenting low-contrast faces to the sensory non-dominant eye while dynamic high-contrast Mondrian patterns were shown to the dominant eye [31]. Participants were required to indicate the target's location as soon as they could perceive it. Accurate localization indicated that faces had entered subjective (modulation of sensory experience) and objective (knowledge of spatial location) awareness [25]. We additionally manipulated the presence of facial information between conditions [27]. Specifically, half of the participants viewed upright faces, while the remaining viewed inverted faces as targets during *b*-CFS. Inverting faces disrupts the perception of specific configurations crucial for identifying human facial structures while preserving basic visual information like luminance and contrast [25,29]. Our *b*-CFS paradigm aimed to distinguish between these high-level (configural) and low-level (visual) influences on the time it takes for faces to 'break suppression.' That is, our setup aimed to distinguish between high-level and low-level influences on breaking times. If lighter faces broke suppression faster than darker faces for upright *and* inverted targets, we could not assert the primacy of high-level or low-level influences. On the other hand, observing breaking time effects exclusive to upright faces would suggest that skin tone influences facial processing only after precise facial configurations have been identified [29].

### Research objective

The primary aim of this investigation was to examine whether colorist effects, operationalized as the differential processing of lighter- and darker-toned non-Melanesian faces, could be inferred across samples of native Melanesian adults. To achieve this, we utilized a range of response measures that systematically constrained participant deliberation opportunities. We explored whether lighter and darker faces, matched for ethnicity, gender, and attractiveness, would yield different outcomes across these measures. The same set of facial stimuli was used across all studies, ensuring that any observed variance in responses was attributable solely to differences in skin tone. The degree of overlap, or a lack thereof, between colorist biases produced under unconstrained, constrained, or perceptual processing conditions will elucidate the extent to which colorist effects might vary with deliberation opportunities [16,17].

Across studies, we tested the null hypothesis $Lighter_d$-$Darker_d \leq 0$, where 'd' represents a specific outcome parameter for each study (detailed under Results). All planned contrasts used Welch's *t*-tests and bias-corrected Hedge's *g* effect size estimates, as these are robust for handling unbalanced samples and violations of the homogeneity of variance assumption [32,33]. Reliably rejecting the null would support the notion that intra-group faces are differentially processed based on skin tone. As all studies pursued the same research question, their methodologies and results are presented collectively.

### Materials and methods

All students enrolled in the Psychology program from September 2021 to July 2022 at the University of the South Pacific were eligible to participate across studies. Specific ethnicity data was not collected in compliance with ethical guidelines. No exclusions were made based on demographic, socioeconomic, or other situational variables, as no hypotheses had been specified concerning these factors. The majority of study participants were incidentally female, reflecting the gender distribution of undergraduate students in the Psychology program. Participant ages ranged between 18 and 38 years. All participants had the option of receiving course credit or a cash reward (FJ$ 10.00) following participation.

**Study 1:** Three hundred and five Melanesian and Polynesian undergraduate Psychology students volunteered for the first study between November 2021 and June 2022. This included 209 Female participants (M = 23.5; SD = 4.6 years), 79 Male participants (M = 26.1; SD = 5.8 years), and 17 Non-binary/other participants (M = 22.6; SD = 4.9 years). In terms of income, 149 Females, 40 Males, and 13 Non-binary/others reported no employment in the past year; 55 Females, 34 Males, and 4 Non-binary/others earned less than 15,000 Fijian dollars ($) annually, while the remaining participants fell into higher income brackets. Sensitivity analyses for paired contrasts, conducted using the *stats* R package [34], indicated our sample could detect small effects ($d_z > 0.16$) with 80% power at a 5% alpha error rate.

**Study 2:** Between December 2021 and June 2022, one hundred sixty-eight undergraduates volunteered for the present study. Data from fifteen participants were excluded due to internet speed issues affecting over 10% of BIAT trials. The remaining one hundred fifty-three participants comprised 109 Female participants (M = 23.3; SD = 4.4 years), 30 Male participants (M = 26.6; SD = 5.8 years), and 1 Non-binary/other participant (23 years). Income distribution was similar to Study 1. 20 Females, and 11 Males reported no employment the previous year, 85 Females, 19 Males, and 1 Non-binary/other participant earned less than $15,000, with the remaining participants reporting earnings between $16,000 and $26,000. Sensitivity analyses for one-sample tests revealed that our sample could detect moderate effects ($d_z > 0.23$) with 80% power at a 5% $\alpha$ error rate.

**Study 3:** Following earlier *b*-CFS sampling conventions [30,31,35], we aimed to recruit sixty participants, thirty for each face orientation (upright/inverted) condition. To be eligible for the study, participants had to report no prior sensitivities to high-contrast visual patterns, the presence of normal/uncorrected vision, and right-handedness. The final cohort consisted of 51 Female participants (M = 24.9; SD = 5.7 years) and 9 Male participants (M = 24; SD = 5.6 years). CFS data from two participants were incomplete due to power outages during data collection (~12% of trials not attempted). CFS data from one participant could not be collected due to a cyclone warning. Income distribution was consistent with the previous studies, with 34 Females and 7 Males reporting no employment, 13 Females and 1 Male reporting earnings less than $15,000, and the remaining participants reporting incomes between $16,000 and $26,000. Sensitivity analyses for paired contrasts indicated our sample could detect moderate-to-large effects ($d_z > 0.53$) with 80% power at a 5% $\alpha$ error rate. Given the prevalence of similar effects and sample sizes in earlier *b*-CFS research [30], a larger sample was not deemed necessary. Nevertheless, we employed Bayes factors to complement our frequentist analyses by assessing the likelihood of our alternative hypothesis regarding colorist bias. All procedures were completed within 40 minutes.

**Face stimuli:** Participants across studies viewed 32 faces comprising 16 males and 16 females. These faces were sourced from the Chicago Face Database, or CFD [14] and contained balanced representations of Asian, Black, Latin, and White groups. Faces with neutral and non-threatening expressions from non-Melanesian groups were selected to minimize potential biases related to emotionality or familiarity [27,36]. Faces were grouped into *High Attractive Female* (*HAF*; AF242, AF255, BF233, BF240, LF243, LF249, WF022, and WF233 CFD-IDs), *High Attractive Male* (*HAM*; AM214, AM216, BM043, BM248, LM201, LM224, WM004, and WM029), *Low Attractive Female* (*LAF*; AF213, AF231, BF007, BF200, LF220, LF253, WF002, and WF210), and *Low Attractive Male* (*LAM*; AM224, AM226, BM213, BM219, LM209, LM240, WM201, and WM206) categories based on attractiveness data provided by [14]. Evaluations for *Low Attractive* (M = 2.15; SE = 0.004) and *High Attractive* (M = 3.45; SE = 0.004) faces provided by our sample mirrors [14]'s classifications, with both pairs of ratings appearing strongly correlated, *r* (30) = .94, *p* < .001. Within each face category (*HAF*, *HAM*, *LAF*, *LAM*), pairs of Asian, Black, Latin, and White faces were split into 'lighter' and 'darker' variants based on relative skin tone, estimated through a multi-step image processing procedure.

**Skin tone estimation:** Skin tone was estimated using the following steps: first, all 2444 by 1718 pixels faces were despeckled, passed through a Gaussian spatial noise filter, rendered achromatic and standardized along contrast levels using the *imager* R package and commercial image processing software (BatchPhoto). Across the processed faces, mean (SD) lumens for darker *HAF*, *HAM*, *LAF*, and *LAM* variants were 0.7 (0.11), 0.62 (0.07), 0.67 (0.14), and 0.68 (0.2), respectively. Mean (SD) lumens for lighter *HAF*, *HAM*, *LAF*, and *LAM* variants were 0.76 (0.03), 0.71 (0.05), 0.73 (0.09), and 0.7 (0.05), respectively. Next, a 339.4 by 76.7 pixels region centered around the forehead was cropped across faces ([Fig 1], Panel C). As all faces had been rendered achromatic, image luminance remained the single variable point estimate across targets. Faces with higher median forehead luminance were categorized as 'lighter' and those with lower luminance were classified as 'darker' within each target category. Forehead

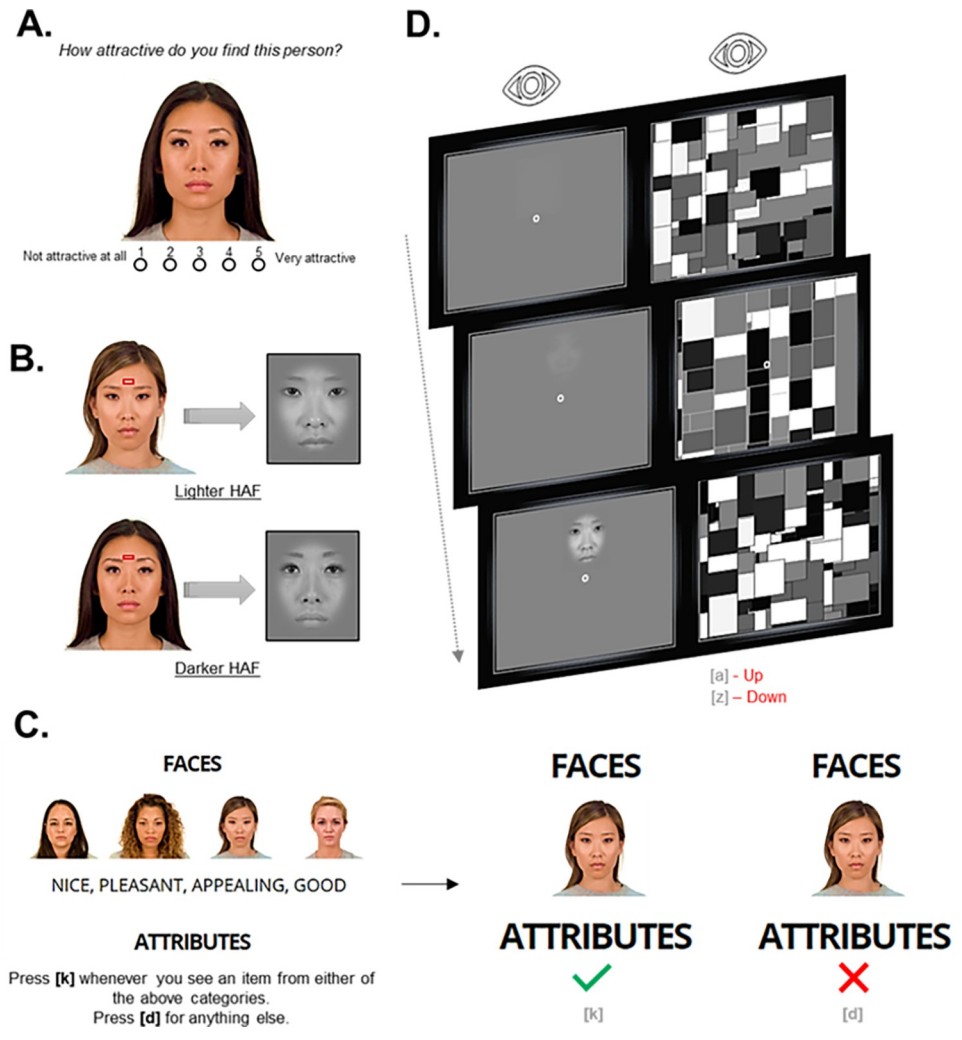

**Fig 1. Examples of face stimuli and tasks used across the three studies.** In Study 1, participants evaluated 32 faces from the Chicago Face Database for attractiveness (Panel A). Faces within each attractiveness-sex-ethnic category were classified into lighter and darker variants based on forehead luminance estimates. Panel B illustrates examples of lighter (CFD-ID: AF255; $Mdn_{Lumens}$ = 0.74) and darker (CFD-ID: AF242; $Mdn_{Lumens}$ = 0.71) highly attractive Asian female targets, with forehead regions from which skin tone was extracted enclosed in red. In Study 2, unprocessed faces were differentially mapped with positive attributes across brief implicit association tests (Panel C). Panel D shows processed faces as they appeared during continuous flash suppression across Study 3. A demo video of the CFS task is available online. Displayed faces are used with permission from Ma et al., 2015 (online).

regions were selected for estimating skin tone as they contained no discernible blemishes/wrinkles/contours across any of the faces sampled. Additionally, [14] reported that forehead regions were perpendicularly located relative to the camera position, meaning they were more likely to be evenly lit and entail minimal refractive variance.

**Study-specific materials:** All unprocessed faces were presented in survey format using Google Forms across **Study 1** and during a brief implicit association test, or BIAT [37] developed on Gorilla [38] across **Study 2**. BIATs additionally presented four positive (NICE, APPEALING, PLEASANT, and GOOD: 25 characters) and four negative (BAD, UNPLEASANT, DISGUSTING, and UGLY: 27 characters) attributes. Working samples of the BIATs are available in the online file.

For **Study 3**, achromatic and processed faces appeared as ovally masked monocular targets during flash suppression. These faces were also presented on laminated 3 by 4-inch cards during the pre-CFS sorting task. For masks, 256 unique Mondrian patterns containing between 50 and 200 randomly sized achromatic rectangles with black, grey, and white hues were generated. Faces and Mondrians were achromatic to reduce spectral leakage between the two displays [25]. The CFS task, developed on E-Prime 3 [39], was administered in a quiet, dark room. Stimuli during CFS appeared on a 24-inch LCD monitor with a 144 Hz refresh rate. The monitor was 20 inches from the principal axis of a fixed chin-and-head rest, with a black wooden divider splitting the enclosed display vertically along the middle. All participants were fitted with customized $+2.5D$ prism lenses, with each lens oriented to one half of the display and their bases facing inward. Stimulus legibility was confirmed by having participants read instructions presented to alternating eyes before CFS onset. Samples of stimuli and a recorded demo of the CFS task are available in the online file. Data organization, analyses, image processing, and manuscript typesetting were completed on the R platform using the *tidyverse* [40], *ggplot2* [41], *ggpubr* [42], *rstatix* [43], *apa* [44], *imager* [45], *kableExtra* [46] and *papaja* [47] packages. Data, task demos, analysis scripts, and the source markdown file are available in the online OSF file.

## Procedure

**Study 1 (Online Surveys):** All participants received an online link containing the information sheet and consent form. After providing consent, participants completed a series of surveys. The first survey required participants to evaluate 32 faces individually along a 5-point Likert scale, ranging from *1—Not attractive at all* to *5—Very attractive* (Fig 1, Panel A). The sequence of faces was randomized for each participant. After providing face evaluations, participants viewed two survey items that explored consciously held intra- and inter-personal skin tone preferences. The first item stated, *Would you rather have*. . . alongside three options (*lighter skin*, *darker skin*, or *your current skin tone*). Next, participants viewed *Would you prefer to have more friends with*. . . alongside three options (*lighter skin*, *darker skin*, or *the same skin tone as you*). Participants then indicated their age, gender, and income level.

Participants also responded to six iterations of the closed-ended question, *Could you see yourself being attracted to a* [Gender-Sex], where 'Gender' was varied between masculine, feminine, and non-binary, and 'Sex' was varied between male and female. Participants could respond to each question using five options (*Definitely*, *I think so*, *Not sure*, *I don't think so*, or *Definitely not*). These surveys were part of an unrelated investigation and had appeared to all participants. As we did not generate any prior hypotheses about the influence of consciously held skin-tone preferences or presumed sexual orientation on any of our outcome parameters, participants were not screened based on these criteria. Surveys were completed in under 30 minutes on average. Participants were instructed to respond using a desktop or laptop if they had access to one.

**Study 2 (Brief implicit association tests):** All participants received an online link with a consent form. After consent, participants were directed to a demographic survey on Google Forms measuring age, gender, income level, and sexual preferences, similar to Study 1. After completing the survey, participants received a second link directing them to the Gorilla platform [38]. Access to the link was restricted to participants using a computer (not a tablet or phone) with a Fiji-specific IP address and a minimum internet connection speed of 4 Mbps to standardize task presentation and minimize timing errors.

Participants who met these criteria underwent four Brief Implicit Association Tests (BIATs) in varied sequences. Each BIAT presented lighter and darker variants from the *HAF*, *HAM*, *LAF*, or *LAM* categories. Variants were matched along gender, race, and attractiveness levels. A Latin square determined the sequence of BIATs, generating 24 sequence permutations randomly distributed across participants. Between BIATs, participants viewed blank, non-timed intervals with the message *Please press the spacebar to continue* (with the task) displayed on the screen. Each BIAT contained four 20-trial blocks per [37]'s recommendations.

At the beginning of a BIAT trial block, participants viewed four target faces near the top of the screen and examples of positive attributes near the bottom. Participants were instructed to press 'k' on their keyboards when they detected a face from the previous display or a positive attribute and to press 'd' for a non-target face or neutral attribute. Across two trial blocks, target and non-target faces consisted of four lighter and four darker variants, respectively. Target and non-target categorizations were reversed across the remaining blocks. During BIAT trials, participants viewed a target/non-target face or a positive/neutral attribute in the center of the screen for 3000 ms. A correct (or incorrect) response within this duration produced a green checkmark (or red x) for 300 ms (Fig 1, Panel C). If no response was detected within 3000 ms, a message stating 'too slow' appeared for 300 ms before the subsequent trial. Test block sequences were randomized between participants and across BIATs. Completion of four BIATs took 30 minutes on average.

**Study 3 (Sorting and Continuous Flash Suppression):** Similar to the first two studies, all participants received an online link with the information sheet and consent form. Following consent and collection of demographic details (age, sex, income level, sexual preference), participants viewed an online calendar on which to schedule a time for coming to the psychology lab. All participants arrived at the laboratory within 48 hours of providing consent. Upon arrival, participants were immediately seated in front of a desk with four decks of cards equidistant from one another. Each deck contained eight achromatic faces printed on individual laminated cards against a grey background. Unknown to participants, decks exclusively had faces from *HAF*, *HAM*, *LAM*, or *LAF* categories. Deck and card placement was randomized between participants.

Participants were instructed to select a deck randomly, then 'spread out the cards' (faces) on the desk and carefully examine them. Near the long edge of the desk, eight 'boxes' were visually demarcated along a row. Participants were instructed to 'move the face (they) liked the most' into the left anchor box, then place relatively less preferred face(s) across the adjacent box(es). In other words, eight faces were ordinally ranked from *most-* to *least* liked. Participants could freely reallocate faces between boxes any number of times and were under no time constraints. Once the participant signaled completion, the experimenter quietly collected the cards and placed them out of sight. The participant was prompted to select from one of the three remaining decks and repeat the task (rank eight faces by subjective preference).

After faces from all four decks had been ranked and collected, participants were moved to a separate desk with the Continuous Flash Suppression (CFS) setup and fitted with prism glasses. The participant's head was stabilized on a chin-and-head rest, with the hands situated near the response keys. Participants were instructed to place their left index and middle fingers

on the 'z' and 'a' letters and the right hand on the spacebar. All responses were collected on a QWERTY low-latency (<2 ms) mechanical keyboard. Letters were taped with up- and down-facing arrowheads to facilitate tactile discrimination. Each eye independently viewed 600 by 800-pixel grey 'boxes' against black backgrounds enclosed by black and white (15 pixels wide) fusion contours throughout the task, with a white fixation circle appearing in the center (of each box). Each participant was instructed to center their eyes near fixation until contours appeared to binocularly converge ('fuse'), at which point the participant was to press the spacebar to begin.

Binocular fusion was reported by most participants within 10 seconds of the starting trial and by all participants within 3 seconds of the second trial. Pressing the spacebar produced a 10 Hz mask (10 Mondrian patterns per second) to one eye. Only the grey box remained visible to the alternate eye for 1800–2000 milliseconds (ms). Next, a low-contrast face emerged 120 pixels above/below fixation to the non-suppressed eye. The face's contrast relative to the grey background was linearly increased for 2000 ms. The maximum contrast target remained on screen for an additional 7000 ms or until a localization response was detected. Participants were instructed to press 'z' (or 'a') as soon as they could detect a face near the display's bottom (or top). Pressing either key replaced both displays with blank grey boxes. During the non-timed inter-trial interval, participants could move their eyes and relax. Before continuing, participants received instructions to achieve binocular fusion before progressing with a spacebar press. All target faces appeared an equal number of times above/below fixation for each eye condition, culminating in 128 trials per participant. Completion of all trials terminated both displays and ended the experiment. Half of all participants underwent *b*-CFS in the presence of upright faces, and the remaining half experienced *b*-CFS in the presence of inverted faces. Procedural and presentation parameters were adapted from earlier b-CFS protocols [29–31].

## Results

### Study 1: Attractiveness ratings

Three hundred and five participants evaluated 32 (16 male, 16 female) emotionally neutral and multiracial faces sourced from the Chicago Face Database [14]. Each face was evaluated using 5-point scales (Fig 1, Panel A). Ratings were grand mean-centered to reduce between-subject variance while retaining level mean differences. Planned contrasts produced reliable evidence for colorist biases in the presence of *HAM* and *LAM* categories (Fig 2, Panel A).

A 2 x 2 x 2 repeated-measures analysis of variance (rANOVA) explored whether mean-centered ratings were influenced by factors such as skin tone, attractiveness levels, and target sex. The ratings appeared normally distributed across $k$ = 6 measurement levels after examining histograms. Violations of sphericity were corrected using the Greenhouse-Geisser method. The three-way interaction was not significant ($p > .27$). Significant two-way interactions were detected between Tone and Sex, $F(1, 304) = 75.45$, $p < .001$, $\eta_p^2 = .20$, and between Sex an Attractiveness, $F(1, 304) = 6.00$, $p = .015$, $\eta_p^2 = .02$. Significant main effects were observed for Tone, $F(1, 304) = 31.90$, $p < .001$, $\eta_p^2 = .09$; for Sex, $F(1, 304) = 160.80$, $p < .001$, $\eta_p^2 = .35$; and for Attractive, $F(1, 304) = 1156.67$, $p < .001$, $\eta_p^2 = .79$.

Planned (lighter vs darker) contrasts across *HAM*, *HAF*, *LAM* and *LAF* categories indicated significantly higher attractiveness ratings in the presence of lighter *HAM*, $t(1219) = 5.94$; $p = .001$; $g[95\%] = 0.17 [0.11, 0.23]$, and lighter *LAM*, $t(1219) = 7.63$; $p = .001$; $g[95\%] = 0.22 [0.16, 0.28]$, compared to their darker counterparts. A statistical difference between lighter and darker *HAF*, $t(1219) = -2.17$; $p = 0.03$; $g[95\%] = -0.06 [-0.12, -0.01]$, did not meet the Bonferroni-adjusted threshold of $p = .0125$. When the data were split by target ethnicity, lighter Black

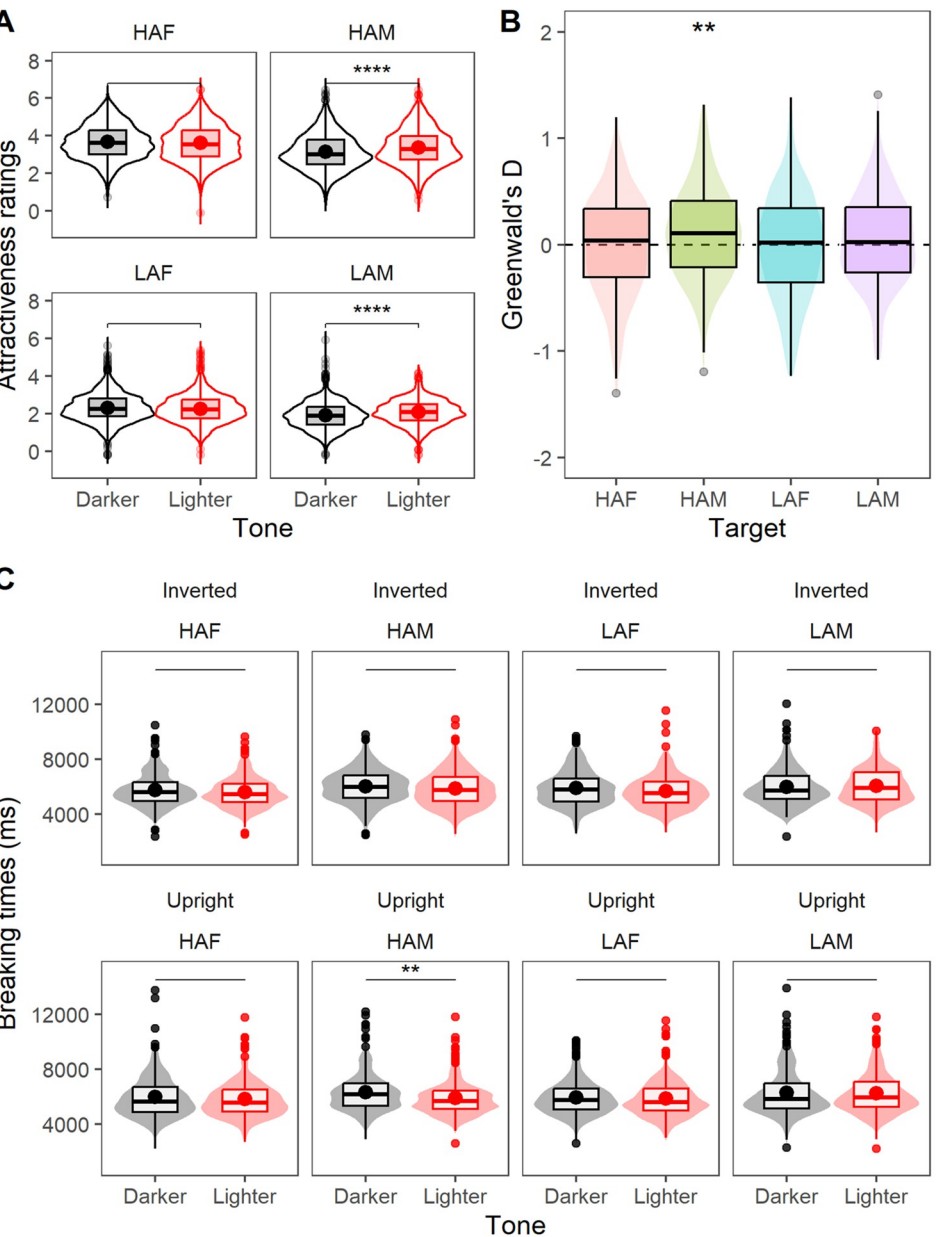

**Fig 2.** Boxplot-violin summaries of mean-centered attractiveness ratings (Panel A), implicit evaluation difference scores (Panel B), and breaking times following continuous flash suppression (Panel C). Asterisks indicate significant (** —$p < .01$; **** —$p < .0001$) differences following two-sample (Panels A, C) and one-sample (Panel B) tests. Facet labels across Panels A and C and $x$-axis ticks across Panel B refer to *High-Attractive Female* (HAF), *High-Attractive Male* (HAM), *Low-Attractive Female* (LAF), and *Low-Attractive Male* (LAM) target categories, respectively. Within each category, 'lighter' and 'darker' variants were equally split along target race.

*HAF*, Asian *HAM*, Black *HAM*, Asian *LAM*, Black *LAM*, and Latin *LAM* were rated as significantly more attractive than their darker counterparts (all $p$'s $\leq .001$, all $g$'s $> .14$); conversely, darker Latin *HAF*, Latin *HAM* and White *LAF* were rated as significantly more attractive than their lighter counterparts (all $p < .01$, all $g$'s $> .11$) [S1].

Target ethnicity (Asian, Black, Latinx, White) was not entered as a factor into our model for two reasons: first, our research focused on *intra*-ethnic group differences along skin tone,

which motivated our stimulus classification strategy. Specifically, we compared lighter and darker variants across targets matched by ethnicity, ensuring any morphometric variation between target races would be averaged out during contrasts. Conversely, including participant race as a factor would accentuate the impact of inter-ethnic morphometric differences, potentially confounding the *intra*-ethnic effects we were interested in. Second, including target ethnicity (or any other variable, such as participant sex) would have quadrupled the number of measurements (from 2 x 2 x 2 to 2 x 2 x 2 x 4), reducing statistical power and increasing model complexity to the point of uninterpretability (48). Nevertheless, for transparency, exploratory 4-way ANOVAs run across $k = 32$ measurement levels have been included in the attached Supporting Information.

Study 1: Explicit skin-tone preferences. After providing facial attractiveness evaluations, participants were asked whether they wanted to change (or retain) their natural skin tone. Out of 209 female and 79 male respondents, 20 female and 5 male respondents wanted darker tones relative to their natural tone. 36 female and 11 male respondents reportedly wanted lighter tones. The remaining 153 female and 63 male respondents wanted to maintain their natural tone. Relative to the total sample, 15% of participants preferred lighter skin tones compared to their natural skin, while 9% preferred darker tones. 9% and 18% of participants preferred friends with lighter or darker skin tones, respectively. A two-sample proportion test indicated that lighter skin was selected significantly more frequently among participants who wanted different skin tones, $\chi^2(1) = 6.22$, $p = .013$.

Participants were also asked whether they would prefer more friends with different/similar tones relative to their own. Of 209 female and 79 male respondents, 35 female and 17 male respondents wanted more friends with darker tones. 16 female and 11 male respondents wanted more friends with lighter tones. The remaining 158 female and 51 male respondents wanted friends with similar skin tones. A second proportion test indicated that among participants who wanted friends with different skin tones, friends of a darker tone were desired significantly more frequently than friends of a lighter tone, $\chi^2(1) = 10.93$, $p < .001$. Next, we explored whether wanting a different skin tone (for oneself or potential friends) was related to each other and facial attractiveness ratings. Given the pronounced imbalance in outcome distributions, the data was resampled (with replacement) using bootstrap to equate the number of observations across different categories of skin tone biases. A Chi-square test of independence revealed significant associations between the categorical predictors, $\chi^2(4) = 810.74$, $p < .001$.

Post hoc analysis of chi-square residuals revealed that participants who preferred a darker skin tone produced residuals of -2.98 and -16.65 when preferring darker- and lighter-skinned friends, respectively, and a residual of 13.11 for showing no friendship bias. Those who preferred a lighter skin tone produced residuals of 2.79 and 4.91 for dark- and light-skinned friend biases, respectively, and a residual of -5.14 for no friendship bias. Participants without a skin tone bias produced residuals of -0.003, 10.56, and -7.05 for preferences towards dark-toned, lighter-toned, and no friendship biases, respectively. A linear model using Ordinary Least Squares (OLS) explored whether facial attractiveness was influenced by the two categorical predictors corresponding to the desire to alter one's skin tone or befriend others with different tones.

The model explained a statistically significant albeit very weak proportion of variance ($R^2 = 3.78e\text{-}04$, $F (8, 31589) = 1.49$, $p = 0.05$, *adj.* $R^2 = 1.25e\text{-}04$). The model produced three significant $\beta$ estimates: first, a preference for lighter-skinned friends was negatively associated with facial attractiveness ratings $\beta_{95\%} = -.11_{-.17, -.04}$; $p = .002$. Second, wanting a lighter skin tone for oneself *and* preferring light-skinned friends, $\beta_{95\%} = .10_{.01, .18}$; $p = .028$, as well as expressing

no preference for one's own skin tone but showing a preference for light-skinned friends, $\beta_{95\%} = -.11_{.02,.19}$; $p = .014$, were both positively associated with facial attractiveness ratings.

## Study 2: Implicit attitudes

One hundred fifty-three participants underwent four Brief Implicit Association Tests (BIATs) that required them to rapidly associate lighter and darker facial variants from *HAF*, *HAM*, *LAF*, or *LAM* categories with positive or neutral attributes alternatively. Across individual trials, participants had to emit keypress responses within 3 seconds of target/attribute onsets, which afforded them less time to deliberate before responding (relative to Study 1).

Standardized differences in mean response times for accurate target-attribute mappings were estimated using [37]'s recommendations. BIAT trials were parsed into two focal categories, Focal Category 1 (FC1) and Focal Category 2 (FC2), based on the block-specific instructions. Across FC1, participants were instructed to map lighter variants with positive attributes and darker variants with neutral attributes. Across FC2, participants were instructed to map darker variants with positive attributes and lighter variants with neutral attributes. Mean (SD) response times for non-focal/incorrect responses (M = 5,522.8, SD = 3058.2 ms) were significantly larger than response times for focal responses (M = 1,044.8, SD = 3058.2 ms; $p < .001$), implying block-specific instructions had been attended to. Next, the first four trials from all trial blocks were dropped to control for practice effects, along with all incorrect responses and trials with keypresses within 200 ms of target onset [37].

Greenwald's difference (*D*) scores were estimated as the ratio of the difference between FC1 and FC2 mean response times for focally correct responses and their inclusive (non-pooled) standard deviation, where Greenwald's $D = \frac{FC1\mu - FC2\mu}{(FC1,FC2)\sigma}$. Scores were computed for individual BIATs (*HAF/HAM/LAF/LAM*). A higher (lower) *D* score implies lighter (darker) faces were mapped more fluently with positive attributes than the contrast category. Levene's test confirmed BIAT response variances were homogenous ($p = 0.21$), with Shapiro tests indicating the data was normally distributed across individual BIATs (all $p$'s > 0.46). A one-way analysis of variance (ANOVA) indicated that Greenwald's *D* did not significantly vary between implementations, $F(3, 600) = 2.13$, $p = .095$, $\eta_p^2 = .01$. Planned one-sample contrasts showed that lighter *HAM* was mapped significantly more frequently with positive attributes relative to darker *HAM*, $t(150) = 2.88$; $p = 0.005$; $g[95\%] = 0.24$ [0.07, 0.4] (Fig 2, Panel B).

## Study 3: Sorting performances

Sixty-one participants sorted eight faces separately, from most (level 1) to least (level 8) preferred, across *HAF*, *HAM*, *LAF*, and *LAM* categories. Before analysis, rankings were parsed into preferred (levels 1–4) and non-preferred (levels 5–8) categories—this parsing strategy aimed to control for response patterns that may have been motivated by ethnic preferences.

To see how, recall that each deck contained equal numbers of lighter and darker variants from each target ethnicity. If participants decided to sort faces based on ethnic characteristics, they would presumably select comparable frequencies of lighter *and* darker variants of the preferred ethnic group over skin-tone variants of the non-preferred ethnic group. And since our skin tone classification (lighter, darker) applied within groups, exhibiting a preference for a particular ethnic group over another would manifest a response distribution indistinguishable from a null estimate. On the other hand, if sorting was influenced by skin tone irrespective of ethnicity, frequencies of lighter and darker variant selections should *collectively* vary between preferred and non-preferred categories. This hypothesis was confirmed following planned two-sample contrasts between preferred and non-preferred selection frequencies. Specifically,

mean frequencies of lighter variant selections across preferred categories were statistically greater for *HAM*, $t(120) = 4.37$; $p = .001$; $g[95\%] = 0.4 [0.21, 0.58]$, *LAM*, $t(119.8) = 5.2$; $p = .001$; $g[95\%] = 0.47 [0.29, 0.66]$, and *LAF* $t(119.7) = 2.63$; $p = 0.01$; $g[95\%] = 0.24 [0.06, 0.42]$.

## Study 3: Breaking times

Across participants who completed *b*-CFS, the first six trials were omitted to control for learning effects. Subsequently, trials with responses recorded within 2000 ms of trial onset, which accounted for 1.2% of total trials, were removed as no perceivable targets could be detected before 2000 ms. Next, all trials with incorrect localization responses were excluded. This left 95.2% and 95.6% of the trials for the upright and inverted face conditions, respectively. Ocular non-dominance was estimated for individual participants following [31]'s suggestions. Specifically, the eye associated with longer breaking times was identified as 'sensory non-dominant' and targeted for suppression [31]. Breaking times for non-dominant relative to dominant eyes were significantly longer across both face conditions (all $p$'s < .001).

Planned contrasts showed significantly faster breaking times for lighter (M = 5912.5, SE = 88.6 ms) relative to darker (M = 6343.4, SE = 102.1 ms) upright *HAM* targets, $t(423.7) = 3.19$; $p = 0.002$; $g[95\%] = 0.15 [0.06, 0.25]$ (Panel C, Fig 2). Contrasts across the remaining categories were non-significant (all $p$'s > 0.09). Bayes factors estimated with .707 Cauchy priors indicated the data was 14 times more likely to occur if the alternative hypothesis of *HAM* colorism was true, $\Delta M = 419.4$, 95% HDI [134.5, 663.3], $BF_{10} = 14$. Supplementary analyses [S2] indicated lighter Asian *HAM*, White *HAM*, and lighter Black *LAM* broke suppression significantly faster than their darker counterparts (all $p$'s < .04; all $g$'s > .21). Frequentist tests and Bayes factors for remaining categories revealed the data was extremely unlikely (all $BF_{10}'s < .8$, all $p$'s > .08) to support the alternative hypothesis of *LAM*, *LAF*, or *HAF* colorism.

## Ad hoc analyses

Performances generated under both unconstrained (surveys, sorting) and constrained (BIAT, *b*-CFS) conditions provided robust evidence for colorist effects (*Lighter*$_d$-*Darker*$_d$ > 0), particularly in the presence of attractive male faces. This focus on male faces was unexpected and prompted further investigation. One feature that stood out was the significant gender imbalance across the studies. Specifically, 69%, 77%, and 87% of participants across Studies 1, 2 and 3 were female, and 26%, 22%, and 15% were male. Inspection of responses to six survey items inquiring about attraction targets (Fig 3) indicated that most male and female respondents were attracted to the opposite sex. The majority of male (female) respondents selected *Definitely* when asked if they could see themselves being attracted to a 'feminine-female' ('masculine-male') and *Definitely not* to all remaining options. While unrelated to our initial hypotheses, these responses suggested that the over-representation of female participants with generally opposite-sex preferences may have contributed to the pattern of *HAM* colorism detected across studies.

To explore the impact of participant gender on outcome parameters, we estimated standardized effect sizes with 1000-bootstrapped 95% confidence intervals (CIs) for all target categories for male and female participants separately (Fig 4). Symmetrical ±.1 equivalence bounds were applied to illustrate statistically non-negligible effects (indicated by CIs exceeding bounds) and/or non-equivalent (CIs that do not overlap with null intercepts—49). Sensitivity analyses were conducted for each gender across each study to estimate the smallest detectable effect with at least 60% power given the sample under consideration. Analyses for non-binary participants were not included due to their small sample size (18 out of 518 participants).

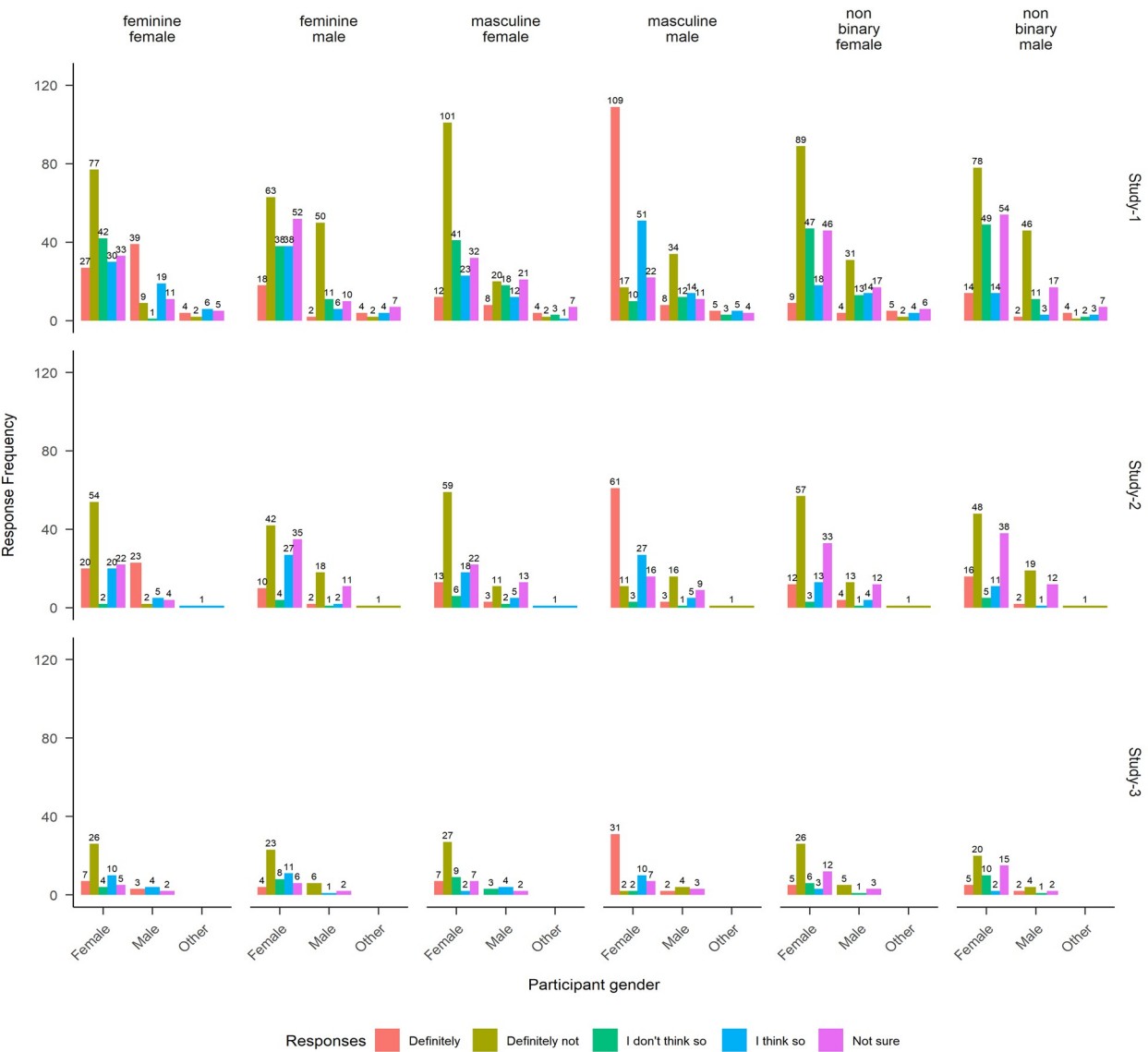

**Fig 3. Response frequencies (*y*-axis) to six iterations of the survey item. *Could you see yourself being attracted to a* [gender-sex] label.** Responses are split by participant gender (*x*-axis). Across studies (rows), males and females primarily responded with "Definitely not" when considering attraction to feminine-males, masculine-females, and non-binary individuals (columns 2, 3, 5, and 6). Male participants predominantly reported a strong attraction to feminine-females, and female participants expressed a similar sentiment towards masculine-males (columns 1 and 4).

Factorial models with participant sex as a factor were not run for two main reasons. First, as previously noted, additional interaction terms would complicate our model significantly [48,49]. Second, the gender imbalance would limit the interpretability of any main effects related to participant sex, even after Type-2 or Type-3 adjustments [50]. Estimating standardized effects for male and female participants separately avoided the complexity and interpretational challenges of a four-way interaction.

Across unconstrained measures (Rows 1 and 5, Fig 4), reliable and non-negligible colorist effects were detected in the presence of *HAM* and *LAM* categories for male and female participants. 16 male participants additionally produced evidence for *LAF* colorism (*d* [95% CI] = 0.91 [0.17, 1.63]) during sorting tests. Across constrained ('implicit') measures (Row 2, Fig 4),

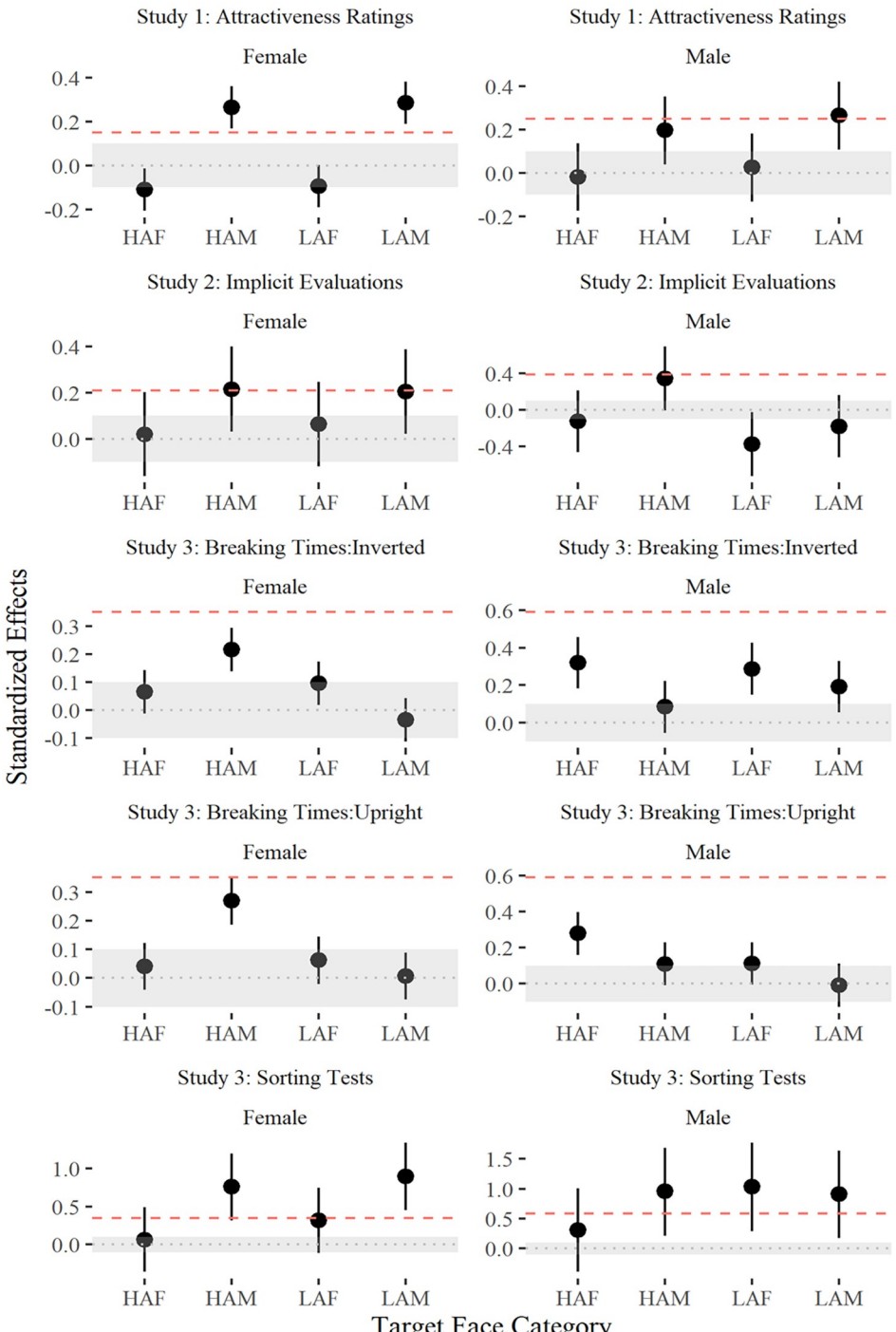

**Fig 4. Standardized effect sizes (*y*-axes) with 95% bootstrapped CIs following *Lighter–Darker* contrasts across measures (panels) for female (left column) and male (right column) participants.** Effects larger than 0 indicate a *Lighter > Darker* variant preference. Shaded regions indicate symmetrical ±.1 equivalence bounds. CIs overlapping equivalence bounds and/or null dotted (grey) intercepts show statistically different and/or equivalent outcomes concerning a negligible effect (-.1 < *d* < .1–49). Dashed (red) intercepts indicate the smallest effect sizes of interest at 60% power following sample-specific sensitivity analyses. Sensitivity thresholds for male respondents were higher as they represent fewer observations (~25% of the total sample identified as male).

34 male and 118 female participants produced non-negligible evidence for implicit *HAM* colorism. Female participants additionally had non-negligible evidence for implicit *LAM* colorism (0.2 [0.02, 0.39]). Inspection of breaking time differences (Rows 3 and 4) indicated lighter *HAM* faces broke suppression faster for female participants exclusively, irrespective of whether targets appeared upright ($N = 20$; 0.27 [0.19, 0.35]) or inverted ($N = 22$; 0.22 [0.14, 0.29]) orientations. Alternatively, male participants detected lighter *HAF* faces non-negligibly faster during upright ($N = 9$; 0.28 [0.16, 0.4]) and inverted ($N = 7$; 0.32 [0.18, 0.46]) orientations. Males only produced non-negligible colorist effects across inverted *LAF* faces (0.29 [0.15, 0.43]).

## Discussion

Across three studies, we examined how Melanesian cohorts processed emotionally neutral non-Melanesian faces varied along relative skin tone after controlling for differences in ethnicity, attractiveness, and gender. Analyses of unconstrained survey responses (Study 1) and sorting performances (Study 3) revealed a preference for lighter over darker male faces, regardless of target attractiveness or the participant's gender. Unconstrained colorist effects were largely non-significant across female faces, except in the sorting tasks across Study 3, where male participants exhibited a significant bias towards lighter, less attractive female (*LAF*) categories. Under constrained conditions in Study 2, male and female respondents displayed colorist biases towards *HAM* categories only, with female respondents additionally showing evidence of automatic *LAM* colorism. Finally, following *b*-CFS in Study 3, lighter and upright *HAM* faces were observed to have broken suppression faster than their darker counterparts, suggesting the former may have been advantaged during perceptual processing. Planned contrasts indicated that our null hypothesis (*Lighter*$_d$-*Darker*$_d \leq 0$) had been reliably rejected in the presence of *HAM* across most outcome measures for male and female respondents alike (see Rows 1, 2, and 5, Fig 4).

One exception to the general trend of *HAM* colorism emerged following inspection of breaking times segmented by participant gender. We observed that lighter *HAM* faces broke suppression significantly faster for female respondents, and lighter *HAF* faces broke suppression faster for male respondents only. Further, these gender-specific effects were consistent across upright and inverted face conditions, suggesting that low-level facial features (e.g., eyebrow thickness, nose widths) may have influenced CFS performances [27]. Across less attractive and/or same-sex faces, skin tone did not appear to privilege perceptual face processing one way or the other. Further, survey responses indicated that most female and male respondents across studies were likely heterosexual (Fig 3), which led us to speculate that observed differences in breaking times might reflect optimized perceptual mechanisms geared toward identifying faces with high reproductive potential. In other words, *HAM* colorist biases among females and *HAF* biases among males seemingly imply that 'skin tone' was conditional to the presence of facial configurations with high reproductive potential *viz* informing attractiveness and opposite-sex characteristics [35,51]. On balance, the male participant pool for Study 3 was notably small, with fewer than six males per condition, which limits representativeness. Additionally, given that *HAM* biases were observed among male and female participants across all response measures susceptible to deliberative influence, it is plausible that both groups were influenced by similar knowledge structures [22]. Perhaps socially constructed beliefs associating lighter skin tones with higher status or perceived privilege became salient when participants could freely deliberate about skin tone before responding, implying male and female Melanesians may hold common colorism-related beliefs [2,16,52]. This claim is indirectly supported by the emergence of colorist biases towards less attractive male (*LAM*) and female (*LAF*) targets when participants were unconstrained from deliberating.

This speculation arose following post-debriefing conversations with some participants, who indicated that they had evaluated certain faces more favorably because they judged the latter representative of disadvantaged social groups [53]. These discussions were not systematically conducted and involved only a random subset of participants, and their mention is purely anecdotal. Any claims about consciously held propositions that could have influenced current outcomes are necessarily speculative since participants' response strategies were not recorded. We did not record participant strategies during data collection to forestall cueing demand characteristics [54]. Similarly, participants were not systematically interviewed about their responding strategies after task completion as these would reflect aggregated recollections susceptible to primacy and recency artifacts [21]. Despite these limitations, future research can attempt to identify the specific evaluation strategies participants employ and identify whether beliefs overlap (or not) between different participant groups to achieve a more comprehensive understanding of the belief structures shaping colorist bias.

Another set of findings pertains to the explicit skin tone preferences recorded across Study 1. Participants had to indicate whether they preferred lighter/darker skin for themselves and potential friends. Most participants preferred maintaining their natural skin tone and sought friends with similar tones. This is not unexpected, seeing how questions about skin tone preferences may be socially sensitive and susceptible to top-down moderation [55]. Across the remaining participants, those who favored a darker skin tone showed no inclination to make friends of different tones. Conversely, those preferring lighter skin for themselves were more likely to choose darker-skinned friends, perhaps motivated by beliefs that they would be perceived as more desirable by contrast [56]. Those without any personal skin tone bias displayed a slight inclination away from dark-skinned friends and a strong preference for lighter-skinned friends. Exploratory analyses with an oversampled dataset indicated that explicit skin tone preferences explained a statistically significant but practically negligible proportion of the variance in facial attractiveness ratings ($R^2 < .001$), suggesting that consciously expressed skin tone attitudes may not directly relate to the evaluative and/or perceptual skin tone preferences detected currently. It is crucial to note that these surveys were not independently validated. They were designed to identify whether colorist biases would be consciously acknowledged when participants were directly questioned about skin tone. Future research should employ non-simulated data and better-validated measures to address these questions more definitively.

We emphasize that all discussions relating to skin tone are specific to intra-group perceptions and cannot be generalized to explain inter-group preferences. Our bifurcation of faces into 'lighter' and 'darker' classifications was made within individual ethnic and gender categories, ensuring any impact of morphometric variations between target genders and/or ethnicities would have been 'averaged out' during planned contrasts. Conversely, if we had incorporated target ethnicity into our analyses, this could exacerbate morphometric-centered differences between ethnic groups. Supporting this claim, supplementary analyses revealed that skin tone effects were broadly consistent across participants and measures, whereas effects associated with target ethnicity varied substantially across participants and measures [S1]. We recommend future research address this concern by pre-screening faces along attractiveness, gender, and ethnicity, as was done here, and controlling for morphometric variability beforehand to discern the influence of the latter precisely. We conclude our discussion after addressing some additional limitations of our design.

## Limitations

One concern applicable across studies may have been the limited number of stimuli used, which may have been inadequate for generating conclusions regarding colorism. We

acknowledge this limitation and recommend that future research expand the stimulus set. Nevertheless, using 32 stimuli can be justified on three grounds. First, we aimed to isolate colorist biases while controlling for other stimulus characteristics that might influence face processing, including attractiveness, gender, and ethnicity [25–27,36]. Thirty-two stimuli allowed for a balanced representation of these variables, organized as Attractiveness (2) x Gender (2) x Ethnicity (4) x Skin Tone (2). Increasing the stimulus set while retaining balance would culminate in 64 targets. This would have doubled participation time and elevated the risk of fatigue-induced response degradation [57], which could have concealed the presently detected effects. Second, maintaining a consistent stimulus set across different measures ensured that structural variations between stimuli did not confound the results. Altering stimuli between studies would obfuscate current interpretations as we could not differentiate between effects motivated by colorist proclivities versus stimulus characteristics. Finally, it can be noted that prior research has used similar numbers of stimuli during CFS [29,56], which we emulated for the sake of consistency, given the preliminary nature of the current investigation. As it stands, our results resonate with [1]'s claims of lighter-toned individuals being evaluated more favorably over their darker-toned counterparts in many Asian communities. We encourage future works to extend the current protocols with different and expanded stimulus sets to identify whether the colorist effects noted presently can be replicated. It would be prudent to deploy region-specific face stimuli to determine whether any of the effects reported presently become significantly affected when participants perceive attractive and opposite-sex faces from one's ethnicity [27,56].

A second criticism could be raised about our use of ovally masked, achromatic faces in Study 3, which may have impeded appraisal of relative skin tones. For instance, according to [51], intra-group attractiveness judgments towards male faces are positively related with the latter's perceived "yellowness", which our achromatic faces could not convey. This design decision may have compromised ecological validity. Our response comprises two parts. First, we wanted to minimize low-level confounds unrelated to skin tone to mitigate extraneous stimulus characteristics influencing CFS performances [25]. Second, outcome patterns recorded across attractiveness ratings (chromatic faces) and sorting tests (achromatic faces), as well as between BIAT (chromatic faces) and *b*-CFS (achromatic faces), were fairly consistent. The overlapping patterns across unconstrained and constrained measures among chromatic and achromatic faces imply that the loss of specific chromatic information did not significantly alter the patterns observed.

Another concern pertains to using Google Forms for data collection, which prevented us from standardizing presentation parameters. This limitation was partly due to our inability to restrict the platform's accessibility to devices with web browsers, and partly due to numerous participants being unable to access personal computers due to movement restrictions at the time of data collection. Given that the quality of web-based survey responses is similar across mobile and PC devices [58], we opted to not exclude participants based on the devices they used to access the survey. Additionally, mean-centered attractiveness ratings (Fig 5) converged toward Gaussian distributions across the majority of cases, which, as per the central limit theorem, would imply the collected data were sufficiently representative of population parameters. Still, we acknowledge that the lack of control over presentation conditions may have been problematic. We recommend that future studies mitigate idiosyncratic contextual factors influencing individual performances to enhance data quality.

A fourth limitation of our study was the inadequate characterization of participants' ethnic and socioeconomic backgrounds. We broadly described our sample as non-White and non-*WEIRD* Melanesians, but this falls short of capturing the differences among Fiji's major ethnic groups, the iTaukei and Indo-Fijians, alongside any other ethnically distinct resident groups

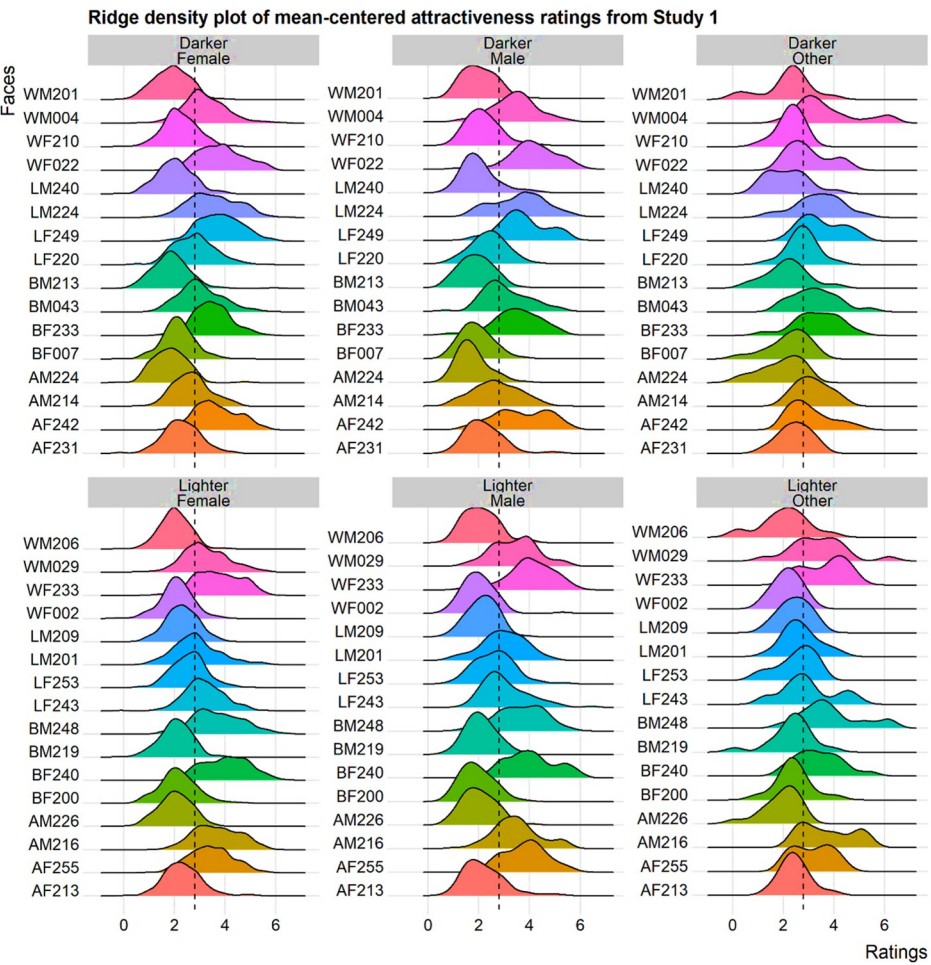

**Fig 5. Ridge density plots of mean-centered ratings (*x*-axes) for individual 'darker' (row 1) and 'lighter' (row 2) faces.** Ratings provided by female, male, and non-binary/other participants are provided across individual columns. The vertical dashed intercept indicates the grand mean ($\mu = 2.80$).

who happened to be sampled (e.g., native Samoans, ni-Vanuatu, Tongans, Kiribati) [59]. If some participants had self-identified with any of our target faces, this would have influenced performances [60,61]. Relatedly, our approximation of socioeconomic status through current income levels failed to account for other socioeconomically important factors like parental income or geographical location. We had not collected information about precise ethnic and socioeconomic backgrounds as our initial hypotheses did not extend to these variables. Consequently, we had not sought ethical approval for collecting ethnic and detailed socioeconomic data, which limited our ability to report along these variables. Recognizing this gap, we recommend that future research in this domain explicitly include such measures, given their potential impact on study outcomes and the need for a more comprehensive understanding.

Next, two concerns regarding our interpretations of *BIAT* effects need to be addressed. First, the scoring convention described by [37] does not differentiate between category-attribute distinctions within trial blocks. This means that a positive *D*-score in Study 2, which was interpreted as evidence of lighter skin bias, could have been generated by faster lighter-positive/darker-negative mappings, slower lighter-negative /darker-positive mappings, or some combination of both. On balance, any of these mappings still aligns with our construct of

interest, 'colorism,' which was operationalized as the preferential processing of lighter- over darker-toned variants. This relates to the second concern, which involves how skin tone can be theorized to have influenced *BIAT* performances, seeing how other stimulus characteristics, such as gender, ethnicity, or attractiveness, might have contributed to the former. In response, recall that lighter and darker skin tone variants were carefully matched on those very characteristics to minimize influences from non-tonal features, which would have been averaged across lighter and darker variants. This lends further credence to the claim that skin tone influences outcome parameters.

Finally, any claims of breaking times reflecting *non-conscious* processing can be contested on methodological grounds [27]. Recall that all participants who underwent flash suppression had to emit speeded localization responses as soon as they consciously detected a face. This approach was informed by earlier work suggesting that localizations mark a target's entry into conscious awareness [25,29,56]. We assumed speeded accurate localizations would capture the "transition points" in a temporally ordered sequence from unconscious to conscious face processing [26]. One concern with this metric is that participants may have varied in their appraisal of the necessary "sensory (subjective) evidence" for response emission. This implies that breaking times could just as well have reflected "*post*-perceptual" effects rather than the moment of transition between non-conscious and conscious awareness (emphasis mine— p. 26). This ambiguity raises questions about whether speeded responses captured the moment faces entered conscious awareness, or whether they reflected a localization response generated after faces had already been consciously identified, undermining any assertions about non-conscious influences [27]. This limitation underscores the need for future research to incorporate non-speeded localization measures supplemented with facial identity checks, as recommended by [26,27]. Such measures will offer a better understanding of the contribution of bottom-up detection and top-down identification during facial processing. On balance, the fact that gender-specific performance differences were exclusive to *b*-CFS suggests that motivational processes underlying face detection/identification among male and female Melanesian participants were sufficiently distinct, irrespective of whether breaking times reflected 'truly unconscious' preferences. Earlier works have already demonstrated that humans tend to automatically allocate attention and processing resources to attractive faces, signaling high reproductive potential [25,29]. Our findings extend on those works by identifying intra-group differences in skin tone as a contributing factor in the processing of attractive, ethnically matched faces from the opposite sex, at least under maximally constrained responding conditions (during *b*-CFS).

## Conclusion

The central aim of our study was to investigate whether colorist biases towards non-Melanesian faces could be detected among Melanesian cohorts as their deliberation opportunities were systematically constrained. Both male and female participants were observed across studies to consistently prefer lighter over darker *HAM* across evaluative measures, suggesting a shared belief system regarding skin tone may have influenced both genders. We also derived preliminary evidence for gender-specific perceptual processing of intra-group faces based on skin tone. Perceptual processing speeds did not statistically vary between lighter and darker variants across any of the less attractive or same-sex face categories, implying skin tone differences were perceptually salient in the presence of attractive and opposite-sex faces. Although we caution against premature extrapolations from a single study, the detection of gender-specific effects at a perceptual (but not evaluative) processing level is worth investigating further. Future extensions can attempt to (dis)confirm the claims made presently while controlling for

the limitations raised earlier. Answering these questions stand to further enrich our understanding of the perceptual and evaluative processes contributing to colorist biases.

## Supporting information

**S1 Table. Exploratory 4-way ANOVA explaining attractiveness as a function of stimulus characteristics.**
(DOCX)

**S2 Table. Exploratory 4-way ANOVA explaining median breaking times as a function of stimulus characteristics.**
(DOCX)

## Acknowledgments

We thank Amitesh Sharma for providing custom prism glasses.

## Author Contributions

**Conceptualization:** Micah Amd.

**Data curation:** Micah Amd.

**Formal analysis:** Micah Amd.

**Funding acquisition:** Micah Amd.

**Investigation:** Micah Amd.

**Methodology:** Micah Amd.

**Project administration:** Micah Amd.

**Software:** Micah Amd.

**Validation:** Micah Amd.

**Visualization:** Micah Amd.

**Writing – original draft:** Micah Amd.

**Writing – review & editing:** Micah Amd.

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
