## [Decision Letter · Decision Letter 0]

3 Aug 2023

PONE-D-23-08925

Intra-group differences in skin tone influence evaluative and perceptual face processing

PLOS ONE

Dear Dr. Amd,

Thank you for submitting your manuscript to PLOS ONE. After careful consideration, we feel that it has merit but does not fully meet PLOS ONE’s publication criteria as it currently stands. Therefore, we invite you to submit a revised version of the manuscript that addresses the points raised during the review process.

Thank you for your valuable submission.

By my own reading, the manuscript still lacks of some details on methods (entire section) and a tightened up Discussion with proper explanation of your findings. You will notice that reviewers found merits in your study, but also raised some *really* important concerns. 

I am not promising we’ll proceed with your study, but, I am keen on authors’ claims and arguments. Importantly, aspects on eligibility, timeframe, workflow, better graphs, better tests, proper CIs and effect sizes etc. are quick to deal, and this is something that authors can solve. The remaining things should be carefully interpreted and addressed. 

Please responde to each comment AND highlight the changes in your text. 

We look forward to receiving your revised manuscript.

Kind regards,

Thiago P. Fernandes, PhD

Academic Editor

PLOS ONE

“MA received an internal research grant from the University of the South Pacific to conduct the current research.”

5. We note that Figure 1 in your submission contain copyrighted images. All PLOS content is published under the Creative Commons Attribution License (CC BY 4.0), which means that the manuscript, images, and Supporting Information files will be freely available online, and any third party is permitted to access, download, copy, distribute, and use these materials in any way, even commercially, with proper attribution. For more information, see our copyright guidelines: http://journals.plos.org/plosone/s/licenses-and-copyright.

Reviewers' comments:

Reviewer's Responses to Questions

**Comments to the Author**

1. Is the manuscript technically sound, and do the data support the conclusions?

Reviewer #1: Partly

Reviewer #2: Yes

Reviewer #3: Yes

2. Has the statistical analysis been performed appropriately and rigorously? 

Reviewer #1: Yes

Reviewer #2: Yes

Reviewer #3: Yes

3. Have the authors made all data underlying the findings in their manuscript fully available?

Reviewer #1: Yes

Reviewer #2: Yes

Reviewer #3: Yes

4. Is the manuscript presented in an intelligible fashion and written in standard English?

Reviewer #1: No

Reviewer #2: Yes

Reviewer #3: No

5. Review Comments to the Author

Reviewer #1: The study aimed to investigate the influence of skin tone on the perception of emotionally neutral faces matched along attractiveness, sex, and ethnicity in Melanesian communities. The study involved three different experiments, which used unconstrained and constrained measures to explore skin tone bias. The results suggested that lighter face preferences were detected for male and some female faces, while implicit colorism was inferred only towards attractive male faces. During continuous flash suppression, lighter and attractive opposite-sex variants broke suppression faster, indicating a bias towards these faces. Overall, the study was well designed and the topic is well-introduced, but some aspects related to the methods, interpretation of results, and underlying rationale require further attention.

Major points:

1. [p. 6, ll. 15-16] I commend the author for investigating colorist biases that have been previously studied in other populations in a Melanesian population. However, why choose a face database composed of non-Melanesian faces? Why ask the population of interest to evaluate faces of other ethnicities? In addition to differences in skin tone, different ethnicities will exhibit considerable morphometric differences that may limit the reliability of the results presented in this study. Please consider carefully explaining this limitation and the rationale behind the choice of face database.

2. [p. 8, ll. 5-6] The imbalance in sample size with respect to gender (209 females, 79 males, and 17 non-binary participants) is given little discussion throughout the manuscript. Some of the main findings, such as a bias for High Attractive Male (HAM) faces, appear to be a consequence of this imbalance. Additionally, it is mentioned in the text, at the end of the general discussion [p. 30, l. 9-10], that the majority of the sample consists of heterosexual women, but the method does not indicate how this data was obtained. Did the participants respond to some kind of scale or was their sexual orientation obtained through open-ended questions?

3. [p. 9, l. 8] In Study 1, attractiveness ratings were collected via links to a Google form. Were the faces evaluated under different screen conditions? Was this controlled for or not? This seems to limit the results concerning attractiveness, as there is a lack of control over the stimulus presentation.

4. [p. 12, ll. 9-17] In this section it is important for the authors to explain why only lighter-skinned men were considered more attractive, as it would be expected to see this effect across all stimulus sets, given the evident colorism in the studied population (as mentioned earlier in the manuscript). I understand that the authors discuss the possibility of participants speculating about the experiment's hypothesis or having ideological motivations. However, in addition to these two explanations, what else could explain why the effect of colorism-influenced evaluation was only found when the facial stimuli were of male individuals? The composition of the sample, for example, is something that should be included in this discussion.

5. The discussion of the first study (as well as the other studies, but especially the first) seems to be limited and does not address other important results that need to be unpacked in the text. For example, what do the results from the explicit skin tone preference form mean in relation to the other results? Do the participants' explicit preferences help explain their attractiveness ratings? This should be carefully included in the discussion section.

6. [p. 26, ll. 21-22] Caution is needed when trying to explain biases related to colorism through the theory of partner selection. This may be interpreted as a scientific defense of the social consequences of colorism. It is important to cite more than one study and restructure the text to avoid this. Ultimately, I suggest removing this passage from the text. Additionally, in [p. 30, ll. 9-11] the text again appears to create non-cautious links between colorist evaluation biases and biological explanations for this phenomenon.

7. Extensive English revision is needed for this manuscript to be published.

Minor points:

1. [p. 4, l. 15] Could you please clarify what is meant by ordinal effects?

2. [p. 6, ll. 12-13] Please note that the figures are placed in a non-intuitive position in the manuscript. Please present the figures whenever they are mentioned for the first time, immediately after the paragraph in which the figure is mentioned.

3. Figure 1: I recommend using a sans-serif font for this figure, as is presented in Figure 2.

4. [p. 6, l. 15] What was the motivation for choosing the number of stimuli? 32 face photographs (16 per gender) seems like a low number of stimuli and should be justified in the text.

5. [p. 6, l. 16] Typo: "Chicago Face Directory". It should actually be referred to as the Chicago Face Database. Reference: Ma, Correll, & Wittenbrink (2015). The Chicago Face Database: A Free Stimulus Set of Faces and Norming Data. Behavior Research Methods, 47, 1122-1135.

6. [p. 10, ll. 12-13] The following variables were included in the ANOVA model: Skin Tone, Target Sex, and Attractiveness Level. Why was the sex of the participant who is evaluating the face not included in the model as well?

7. [p. 15, ll. 1-2] What was the purpose of including these other surveys since they are not related to the tested hypothesis?

8. For all three studies, it is important to describe the socioeconomic variables of the sample that may be biasing the results of the present research. It is also necessary to describe the sample for each study. Perhaps even checking if there are statistically significant differences between the different samples to confirm that this is not a possible confounding factor.

9. [p. 19, ll. 20-21] As soon as the final sample size is presented, please include all the information that describes the sample. In the current version of the manuscript, the information is a bit diffuse, making it difficult to understand.

10. [p. 21, ll. 6-20] The purpose of using the Sorting test in Study 3 is not clear. Only in the results section does the motivation for using this test become clear. I suggest explaining the purpose of the test in relation to the hypothesis as soon as the test is presented.

11. [p. 27, ll. 15-17] Which interviews are the authors referring to? Why was this not mentioned in the method? And why were these results not presented first in the appropriate section?

Reviewer #2: Peer-review report

The manuscript presents three studies that investigate whether individuals from Melanesian communities exhibit colourist biases, specifically a preference for lighter skin tones regardless of factors such as sex, attractiveness, and others. To explore this question, the authors employed ordinal evaluations in Study 1, implicit association tests in Study 2, and the breaking Continuous Flash Suppression (bCFS) procedure in Study 3. The manuscript is well-written, and the studies are meticulously conducted. The results obtained across the studies are consistent, and the discussion sections effectively strike a balance between drawing conclusions and acknowledging limitations. Furthermore, the authors have conducted numerous exploratory and supplementary analyses, and their interpretation of the findings is both convincing and articulate. I do have a few concerns, however:

1. The authors provide a solid rationale for conducting Study 3, which utilises the bCFS procedure, as they argue that previous findings from their other studies might be influenced by spurious propositional factors compared to remaining evaluative measures. While their reasoning seems reasonable, recent developments in the field have raised doubts about the validity of the bCFS procedure, particularly in attributing differences in breakthrough times to high-level unconscious processing (Lanfranco et al., 2023b; Pournaghdali & Schwartz, 2020). In this context, the review paper by Lanfranco et al. (2023) becomes particularly relevant as it discusses the limitations of the bCFS procedure in the context of face processing. Although the authors touch upon concerns about potential low-level visual confounds in their discussion, their treatment of the topic appears somewhat superficial. It is worth noting that more stringent CFS procedures, such as non-speeded accuracy-based tasks (Lanfranco et al., 2022, 2023a; Stein, 2019; Stein & Peelen, 2021), could have been employed. Considering these recent advancements, the authors should clarify why they opted for the bCFS procedure or, at the very least, suggest the potential improvements that future studies utilising non-speeded accuracy-based tasks could bring.

2. The authors demonstrate meticulousness in their experimental designs and framing of statistical analyses. However, I have concerns regarding their approach to processing face images in Study 3. Specifically, they used achromatic faces, which, although intended to control for low-level confounds to some extent, raises questions about the effectiveness of these stimuli in manipulating their variable of interest. The authors have made commendable efforts to address these matters, but they should engage in a more thorough discussion. For instance, they introduced a face-orientation manipulation, which warrants more extensive exploration. The authors should discuss this manipulation in greater detail, as it allows for testing whether breakthrough times are influenced by low-level features rather than high-level configural features. Moreover, this manipulation is relevant to each stimulus category. To address these concerns more effectively, the authors could consider running the experiment with a sample of participants who do not exhibit any indication of colourist bias, as determined by methods similar to those used in Studies 1 and 2. Although I do not expect the authors to conduct an entirely new experiment with a completely different group of participants, I encourage them to expand their discussion to include these concerns.

3. The authors attribute a preference for male over female faces to their effects of interest. While they mention an opposite-sex skin tone effect in the general discussion, it appears that the most prominent and reliable effects observed relate to HAM and LAM colourism. I believe my interpretation is accurate, but I would appreciate clarification from the authors. If my understanding is correct, do the authors perceive a symmetry in these effects? In other words, do they believe that colourist biases are more pronounced for opposite-sex faces? Additionally, did the authors inquire about participants' sexual orientation? Is there a way to examine whether this opposite-sex effect is contingent upon the respondents' sexual orientation?

Now, addressing some minor corrections (e.g., typos):

4. On page 15, lines 9 and 10, the authors refer to mean reaction times, but these should be referred to as response times. Additionally, they should include the standard deviations alongside the means.

5. On page 16, line 10, the word "differed" has been omitted.

6. On page 16, line 15, the phrase "to earlier" is unclear and requires further clarification.

7. On page 17, line 12, the contraction "it's" should be replaced with the possessive form "its."

8. On page 20, line 4, when reporting the age range of the participants, the authors should also provide the mean age, standard deviation, and age range.

9. On page 28, line 15, it says “who happened to sampled”. Please, correct.

10. On page 28, line 21, there is a point that should be a comma.

In summary, this manuscript, consisting of three studies, addresses an intriguing question of social relevance. The research design is robust, the analyses are rigorous, and the discussion of the findings is persuasive and articulate. I am prepared to support the publication of this manuscript once the authors have addressed the concerns I have raised.

References

Lanfranco, R. C., Rabagliati, H., & Carmel, D. (2023a). Assessing the influence of emotional expressions on perceptual sensitivity to faces overcoming interocular suppression. Emotion, 1–21. https://doi.org/10.1037/emo0001215

Lanfranco, R. C., Rabagliati, H., & Carmel, D. (2023b). The importance of awareness in face processing: A critical review of interocular suppression studies. Behavioural Brain Research, 437, 114116. https://doi.org/10.1016/j.bbr.2022.114116

Lanfranco, R. C., Stein, T., Rabagliati, H., & Carmel, D. (2022). Gaze direction and face orientation modulate perceptual sensitivity to faces under interocular suppression. Scientific Reports, 12(1), Article 1. https://doi.org/10.1038/s41598-022-11717-4

Pournaghdali, A., & Schwartz, B. L. (2020). Continuous flash suppression: Known and unknowns. Psychonomic Bulletin & Review, 27(6), 1071–1103. https://doi.org/10.3758/s13423-020-01771-2

Stein, T. (2019). The Breaking Continuous Flash Suppression Paradigm: Review, evaluation, and outlook. In G. Hesselmann (Ed.), Transitions between Consciousness and Unconsciousness (pp. 1–38). Routledge. https://www.taylorfrancis.com/chapters/breaking-continuous-flash-suppression-paradigm-timo-stein/10.4324/9780429469688-1

Stein, T., & Peelen, M. V. (2021). Dissociating conscious and unconscious influences on visual detection effects. Nature Human Behaviour, 5(5), 612–624. https://doi.org/10.1038/s41562-020-01004-5

Reviewer #3: Thank you for the opportunity to review the manuscript.

I believe it has considerable scientific importance with great potential for publication. However, the way the text is currently organized makes it difficult to understand the work.

Thus, I provide below comments that I consider important to improve the work, assisting in its final publication.

Mainly because it is three studies in one, the way this information is presented generates some confusion. In the abstract, for example, I missed the methodological aspects and the conclusion of the study. I believe that the main work of the authors will be the construction of topics such as introduction, methodology, discussion and conclusion in a way that brings together the information of the work in a general way.

I believe that the keywords chosen do not reflect well the main objective of the study. I suggest that the authors could re-evaluate.

In the introduction, I missed previous studies that talk about the topic, demonstrating the importance of the study being conducted. Also, the introduction is confusing. It starts by talking about the topic, then addresses topics of the methodology used, then goes back to the objective of the study. My suggestion is: to make a general introduction about the study carried out, starting from the most general aspects to the most specific (citing previous studies, for example), ending with the objective and hypotheses of the study, being able to inform in a very brief way that to achieve these objectives three studies were carried out.

All methodological aspects are better described in the methodology section.

Regarding the methodology, I believe that the authors can also carry out a general methodology with subtopics for the methodology used in each study carried out. I believe that the inclusion of information on these methodological aspects is very important, such as the inclusion and exclusion criteria of the sample, the period of data collection, how it was carried out online, what care the researchers used to avoid response bias in filling out these forms, among others.

Better describe the analyses performed in each study, which variables were being evaluated, whether sample calculation was performed, whether the assumptions for performing the tests were met, etc.

In my opinion, a general discussion uniting the characteristics of the three studies is more interesting than three discussions and a general discussion at the end. Mainly because in the specific discussion of the studies, the authors are only resuming the results and are not bringing the studies that corroborate or refute these results. Another tip for the general discussion would be to resume the objective and hypotheses of the study, to speak briefly and without reporting the numerical data again, what the main results found, and to discuss these results based on the literature.

In the conclusion, I suggest that the authors can provide the conclusions of the study, without using references. It is the moment that the researcher can bring his perspectives on what was found and what remains to be researched, providing suggestions for future research.

Sincerely.

6. PLOS authors have the option to publish the peer review history of their article (what does this mean?). If published, this will include your full peer review and any attached files.

Reviewer #1: No

Reviewer #2: No

Reviewer #3: No

---

## [Author Response · Author response to Decision Letter 0]

6 Nov 2023

We require you to either (1) present written permission from the copyright holder to publish these figures specifically under the CC BY 4.0 license, or (2) remove the figures from your submission

Written permission from the copyright holder has been acquired in relation to the specific figure under question. This has been attached with the Supporting Information.

Editor comments:

By my own reading, the manuscript still lacks of some details on methods (entire section) and a tightened up Discussion with proper explanation of your findings. You will notice that reviewers found merits in your study, but also raised some *really* important concerns. 

I am not promising we’ll proceed with your study, but, I am keen on authors’ claims and arguments. Importantly, aspects on eligibility, timeframe, workflow, better graphs, better tests, proper CIs and effect sizes etc. are quick to deal, and this is something that authors can solve. The remaining things should be carefully interpreted and addressed. 

 Please respond to each comment AND highlight the changes in your text. 

Thank you very much for the opportunity to revise the manuscript. All of the reviewers’ points have been individually addressed. The relevant changes have been highlighted in a marked version of the revised manuscript. 

Reviewer 1 comments:

The study aimed to investigate the influence of skin tone on the perception of emotionally neutral faces matched along attractiveness, sex, and ethnicity in Melanesian communities. The study involved three different experiments, which used unconstrained and constrained measures to explore skin tone bias. The results suggested that lighter face preferences were detected for male and some female faces, while implicit colorism was inferred only towards attractive male faces. During continuous flash suppression, lighter and attractive opposite-sex variants broke suppression faster, indicating a bias towards these faces. Overall, the study was well designed and the topic is well-introduced, but some aspects related to the methods, interpretation of results, and underlying rationale require further attention.

We are grateful for your constructive suggestions, and hope the revisions are to your satisfaction

Major points:

1. [p. 6, ll. 15-16] I commend the author for investigating colorist biases that have been previously studied in other populations in a Melanesian population. However, why choose a face database composed of non-Melanesian faces? Why ask the population of interest to evaluate faces of other ethnicities? In addition to differences in skin tone, different ethnicities will exhibit considerable morphometric differences that may limit the reliability of the results presented in this study. Please consider carefully explaining this limitation and the rationale behind the choice of face database.

The choice to use a non-Melanesian face database was driven by two factors: first, using Melanesian faces could have introduced confounding factors related to racial biases (due to the presence of own-race faces), overshadowing the focus on colorist attitudes. Second, we did not have access to a validated database of Melanesian faces controlled for age, attractiveness, and emotionality [p. 4, l.21 – p. 5, l. 9]. On p. 34, l. 2-13, we note that skin tone categories were standardized within specific ethnic and gender groups to minimize morphometric variations. Incorporating ethnicity into our analyses could have amplified differences between ethnic groups, as shown by our supplementary analyses where skin tone effects were consistent, but those linked to ethnicity were not. We suggest future studies control for morphometric variability by pre-screening for attractiveness, gender, and ethnicity to accurately assess the impact of these factors.

2. [p. 8, ll. 5-6] The imbalance in sample size with respect to gender (209 females, 79 males, and 17 non-binary participants) is given little discussion throughout the manuscript. Some of the main findings, such as a bias for High Attractive Male (HAM) faces, appear to be a consequence of this imbalance. Additionally, it is mentioned in the text, at the end of the general discussion [p. 30, l. 9-10], that the majority of the sample consists of heterosexual women, but the method does not indicate how this data was obtained. Did the participants respond to some kind of scale or was their sexual orientation obtained through open-ended questions?

We acknowledge the reviewer's concern about the gender imbalance in our sample and have updated our manuscript to highlight this on [p. 9, l. 3-4; p. 26, l. 1-8] . The imbalance was unintentional and roughly represents gender distributions across student cohorts. We elaborate on how this could have influenced the broad bias for High Attractive Male (HAM) faces. The gender balance is used to justify effect estimations for participant genders separately. Sexual orientation was inferred from responses to six closed-ended questions about attraction to various gender-sex labels that were part of the demographic surveys. We have summarized the survey responses in a new Figure 3 in the revised manuscript. We clarify that all attractiveness-unrelated surveys were part of a separate investigation which was only included for additional insight in relation to our findings [described on p. 13, l. 14-23].

3. [p. 9, l. 8] In Study 1, attractiveness ratings were collected via links to a Google form. Were the faces evaluated under different screen conditions? Was this controlled for or not? This seems to limit the results concerning attractiveness, as there is a lack of control over the stimulus presentation.

We acknowledge the reviewer's concern about the lack of control over screen conditions for collecting attractiveness ratings in Study 1, under the updated Limitations section [p. 35, l. 9-20]. Due to movement restrictions at the time of data collection, many participants could not attend university, and while they were instructed to use a desktop or laptop, this could not be verified from our side given the limitations of the platform. Despite these limitations, the mean-centered attractiveness ratings across individual faces converged towards Gaussian distributions, indicating that recorded outcomes were representative of population parameters (Figure 4). Additionally, (Mockel et al 2015) demonstrated that survey responses may not significantly vary between mobile and PC users in terms of quality. We recommend future studies should aim to standardize presentation conditions to enhance data quality.

4. [p. 12, ll. 9-17] In this section it is important for the authors to explain why only lighter-skinned men were considered more attractive, as it would be expected to see this effect across all stimulus sets, given the evident colorism in the studied population (as mentioned earlier in the manuscript). I understand that the authors discuss the possibility of participants speculating about the experiment's hypothesis or having ideological motivations. However, in addition to these two explanations, what else could explain why the effect of colorism-influenced evaluation was only found when the facial stimuli were of male individuals? The composition of the sample, for example, is something that should be included in this discussion.

We have provided an overview of our sample’s composition under the Participants section [pp. 9-10]. In our updated Discussion on we speculate that the importance of skin tone may have been related to the presence of facial features that signal reproductive value, such as attractiveness and opposite-sex characteristics [p. 31, l. 2-11]. Our sample's gender composition, which predominantly comprised of heterosexual females, was theorized to have influenced this outcome. Our inferences about sexual orientation were derived from a six-item survey that showed a strong preference for 'masculine males' across the majority of female respondents. However, we also note on [p.38] that consistent HAM colorist biases were detected across predominantly heterosexual male respondents, at least across evaluative measures. This finding contradicts our earlier speculation that skin tone serves as a salient cue only in the context of attractive and opposite-sex faces, at least at an evaluative level of processing. Coupled with the observation that colorist biases were more pronounced under unconstrained responding conditions, we speculate that male and female participants may have been motivated by common (unknown) belief structures [p.38, l. 11-16].

5. The discussion of the first study (as well as the other studies, but especially the first) seems to be limited and does not address other important results that need to be unpacked in the text. For example, what do the results from the explicit skin tone preference form mean in relation to the other results? Do the participants' explicit preferences help explain their attractiveness ratings? This should be carefully included in the discussion section.

In response to the reviewer's suggestion to elaborate on the relationship between explicit skin tone preferences and attractiveness ratings, we have expanded our Results on [p. 22, l. 1-9], as well as our Discussion on [pp. 32-33,]. Most participants reported no explicit skin-tone preferences., which is not unexpected since responses to socially sensitive questions are likely to be subjectively moderated. To identify whether observed patterns differentially predicted attractiveness ratings, we over-sampled the data to contrive a balanced though artificial dataset. An exploratory OLS regression indicated that explicit skin tone preferences significantly explained less than 0.01% of the variance across attractiveness ratings. Skin tone preferences, at least as currently measured, was not a strong predictor of attractiveness evaluations. We recommend that future research to employ non-simulated data to more definitively address these questions.

6. [p. 26, ll. 21-22] Caution is needed when trying to explain biases related to colorism through the theory of partner selection. This may be interpreted as a scientific defense of the social consequences of colorism. It is important to cite more than one study and restructure the text to avoid this. Ultimately, I suggest removing this passage from the text. Additionally, in [p. 30, ll. 9-11] the text again appears to create non-cautious links between colorist evaluation biases and biological explanations for this phenomenon.

We appreciate the cautionary note on the potential implications of linking colorism to the theory of partner selection. We have removed the passages in question and thoroughly revised our manuscript to avoid any interpretation that could be construed as a scientific defense of the social consequences of colorism. We further emphasize on that our speculations are confined to intra-group perceptual processes that cannot be generalized to inter-group dynamics [p. 33, l. 1-13]. 

7. Extensive English revision is needed for this manuscript to be published.

The manuscript has been carefully proofread and extensively revised to meet academic standards for clarity and precision.

===

Minor points:

1. [p. 4, l. 15] Could you please clarify what is meant by ordinal effects? – By "ordinal," we imply relatively ranked categories that are subjectively perceived as being more or less positively valenced relative to each other. This is provided on [p. 5, l. 15-17]. 

2. [p. 6, ll. 12-13] Please note that the figures are placed in a non-intuitive position in the manuscript. Please present the figures whenever they are mentioned for the first time, immediately after the paragraph in which the figure is mentioned. – All figures are provided immediately after the paragraph where they are first mentioned.

3. Figure 1: I recommend using a sans-serif font for this figure, as is presented in Figure 2.– Figure 1 now uses a similar sans-serif font

4. [p. 6, l. 15] What was the motivation for choosing the number of stimuli? 32 face photographs (16 per gender) seems like a low number of stimuli and should be justified in the text. - We acknowledge that 32 stimuli may have been too restricted under Limitations, and justify our choice on three grounds [p. 33, l. 15 – p. 34, l. 13]. First, our study aligns with prior CFS research that used similar numbers of stimuli. Second, the 32 stimuli allowed for a balanced representation of target attractiveness, gender, ethnicity, and skin tone, as organized in a 2x2x4x2 design. Increasing the stimulus set to 64 would have doubled the participation time, elevating the risk of response fatigue. Finally, by keeping stimulus sets consistent, we controlled for structural differences between stimuli from influencing outcomes. 

5. [p. 6, l. 16] Typo: "Chicago Face Directory". It should actually be referred to as the Chicago Face Database. Reference: Ma, Correll, & Wittenbrink (2015). The Chicago Face Database: A Free Stimulus Set of Faces and Norming Data. Behavior Research Methods, 47, 1122-1135. .– Corrected

6. [p. 10, ll. 12-13] The following variables were included in the ANOVA model: Skin Tone, Target Sex, and Attractiveness Level. Why was the sex of the participant who is evaluating the face not included in the model as well? - We outline two reasons for excluding participant sex as a factor on [p. 26, Footnote 2]. First, adding a fourth interaction term would have significantly complicated our model and subsequent interpretations. Second, our sample had a pronounced gender imbalance, which would have limited the interpretability of any main effects related to participant sex, even after applying Type-2 or Type-3 adjustments. The decision to estimate standardized effects for male and female participants separately avoided the complexity and interpretational challenges of a four-way interaction. We emphasize that executing Welch’s tests and using bias-corrected effects for male and female participants independently is a more statistically sound and defensible approach compared to incorporating ‘participant sex’ into a factorial model as an additional predictor.

7. [p. 15, ll. 1-2] What was the purpose of including these other surveys since they are not related to the tested hypothesis? – We mention on [p. 13, l. 18-21] that these surveys were part of an unrelated investigation and were completed by all participants. Although unrelated to our initial hypotheses, survey responses were inspected for additional insight after planned analyses. This revealed that the majority of female respondents may have held opposite-sex preferences, which motivated our ad hoc analysis section. We stress that these surveys have not been independently validated and serve largely an exploratory role to inform subsequent investigations.

8. For all three studies, it is important to describe the socioeconomic variables of the sample that may be biasing the results of the present research. It is also necessary to describe the sample for each study. Perhaps even checking if there are statistically significant differences between the different samples to confirm that this is not a possible confounding factor. – We acknowledge the reviewer's concern about the lack of detailed socioeconomic variables in our sample on [p. 37, l.7-14]. Our approximation of socioeconomic status was primarily through current income levels, which falls short of capturing other socioeconomically important factors like parental income or geographical location. We recommend future works to incorporate these variables for a more nuanced understanding. Our sample was largely consistent in terms of education level, as they were all registered psychology undergraduate students. Demographic indicators, such as age and sex distributions, were similar throughout studies [pp. 9-10]. This suggests that sample characteristics were sufficiently consistent to not act as a confounding factor during analyses.

9. [p. 19, ll. 20-21] As soon as the final sample size is presented, please include all the information that describes the sample. In the current version of the manuscript, the information is a bit diffuse, making it difficult to understand. – All information regarding our sample(s) have been provided under a single section [pp. 9-11].

10. [p. 21, ll. 6-20] The purpose of using the Sorting test in Study 3 is not clear. Only in the results section does the motivation for using this test become clear. I suggest explaining the purpose of the test in relation to the hypothesis as soon as the test is presented. – We clarify on [p. 5, l. 14-17] that sorting tests were included to “reliably distinguish” between differentially valenced stimulus categories. The test was designed to allow participants to ordinally sort faces based on their subjective preferences across four categories: HAF, HAM, LAF, and LAM. Administered under 'unconstrained' conditions, the test controlled for potential biases in rankings that could be motivated by target ethnicity. This is elaborated on [p. 24, l.6-12]

11. [p. 27, ll. 15-17] Which interviews are the authors referring to? Why was this not mentioned in the method? And why were these results not presented first in the appropriate section? – We clarify on [p. 37, l. 4-13] that these discussions were not systematically conducted and involved only a random subset of participants. These were informal sessions centered around participants’ subjective evaluations of the task on mostly procedural grounds and were not directly related to any of our task predictions. Their mention is purely anecdotal. We emphasize that any claims about motivating belief structures are speculative as we did not record subjective response strategies presently.

Reviewer 2 comments:

The manuscript presents three studies that investigate whether individuals from Melanesian communities exhibit colourist biases, specifically a preference for lighter skin tones regardless of factors such as sex, attractiveness, and others. To explore this question, the authors employed ordinal evaluations in Study 1, implicit association tests in Study 2, and the breaking Continuous Flash Suppression (bCFS) procedure in Study 3. The manuscript is well-written, and the studies are meticulously conducted. The results obtained across the studies are consistent, and the discussion sections effectively strike a balance between drawing conclusions and acknowledging limitations. 

We are grateful for your generous evaluation. We hope the revisions have adequately addressed your concerns.

Furthermore, the authors have conducted numerous exploratory and supplementary analyses, and their interpretation of the findings is both convincing and articulate. I do have a few concerns, however:

1. The authors provide a solid rationale for conducting Study 3, which utilises the bCFS procedure, as they argue that previous findings from their other studies might be influenced by spurious propositional factors compared to remaining evaluative measures. While their reasoning seems reasonable, recent developments in the field have raised doubts about the validity of the bCFS procedure, particularly in attributing differences in breakthrough times to high-level unconscious processing (Lanfranco et al., 2023b; Pournaghdali & Schwartz, 2020). In this context, the review paper by Lanfranco et al. (2023) becomes particularly relevant as it discusses the limitations of the bCFS procedure in the context of face processing. Although the authors touch upon concerns about potential low-level visual confounds in their discussion, their treatment of the topic appears somewhat superficial. It is worth noting that more stringent CFS procedures, such as non-speeded accuracy-based tasks (Lanfranco et al., 2022, 2023a; Stein, 2019; Stein & Peelen, 2021), could have been employed. Considering these recent advancements, the authors should clarify why they opted for the bCFS procedure or, at the very least, suggest the potential improvements that future studies utilising non-speeded accuracy-based tasks could bring.

We are grateful for the valuable references. These have been incorporated into the revised manuscript. We clarify that our b-CFS protocol was adapted from previous designs to maintain consistency [p. 6, l. 20]. We acknowledge the limitations of this b-CFS procedure with respect to speeded localization responses on [p. 38, l. 4-24]. We re-iterate Lanfranco’s (2023) claim that speeded localization responses cannot differentiate between (top-down) identification and (bottom-up) detection responses. We advocate for future studies to compare between CFS protocols to better differentiate between perceptual and top-down influences. For balance, we mention how gender-specific performance differences were exclusive to b-CFS, implying the motivational processes underlying face detection/identification among male and female Melanesian participants were sufficiently distinct, irrespective of whether breaking times reflected ‘truly unconscious’ preferences.

2. The authors demonstrate meticulousness in their experimental designs and framing of statistical analyses. However, I have concerns regarding their approach to processing face images in Study 3. Specifically, they used achromatic faces, which, although intended to control for low-level confounds to some extent, raises questions about the effectiveness of these stimuli in manipulating their variable of interest. The authors have made commendable efforts to address these matters, but they should engage in a more thorough discussion. For instance, they introduced a face-orientation manipulation, which warrants more extensive exploration. The authors should discuss this manipulation in greater detail, as it allows for testing whether breakthrough times are influenced by low-level features rather than high-level configural features. Moreover, this manipulation is relevant to each stimulus category. To address these concerns more effectively, the authors could consider running the experiment with a sample of participants who do not exhibit any indication of colourist bias, as determined by methods similar to those used in Studies 1 and 2. Although I do not expect the authors to conduct an entirely new experiment with a completely different group of participants, I encourage them to expand their discussion to include these concerns.

The rationale for the face-orientation manipulation is described in more detail on [p.7, l. 3-13]. We clarify that inverting faces disrupts holistic or configural processing while retaining 'low-level' elemental information. We note that, when participants were split along gender, breaking time differences were observed across upright and inverted faces, implying low-level influences may not be discounted [p. 30, l. 20 - 22]. This led us to speculate that breaking times might reflect optimized perceptual mechanisms geared toward identifying faces with high reproductive potential [p. 31, l. 2-7]. We have incorporated the reviewer's suggestion for future research to pre-screen participants based on colorist biases [p. 33, l. 9-12]. We’ve responded to the concerns about our usage of ovally masked achromatic faces on [p. 34, l.14-23]. Our choice to use ovally masked achromatic faces in Study 3 aimed to control for low-level visual features, which may have reduced ecological validity. However, since outcome patterns recorded across attractiveness ratings (chromatic faces) and sorting tests (achromatic faces), as well as between implicit tests (chromatic faces) and b-CFS (achromatic faces), were fairly consistent, the loss of specific chromatic information did not appear to significantly influence outcome parameters. 

3. The authors attribute a preference for male over female faces to their effects of interest. While they mention an opposite-sex skin tone effect in the general discussion, it appears that the most prominent and reliable effects observed relate to HAM and LAM colourism. I believe my interpretation is accurate, but I would appreciate clarification from the authors. If my understanding is correct, do the authors perceive a symmetry in these effects? In other words, do they believe that colourist biases are more pronounced for opposite-sex faces? Additionally, did the authors inquire about participants' sexual orientation? Is there a way to examine whether this opposite-sex effect is contingent upon the respondents' sexual orientation?

The reviewer correctly observes that the most prominent effects relate to HAM and LAM colorism. We elaborate these points on [p. 31, l. 1-7; p. 34, l. 11-13; p. 39, l. 3-4]. We mention that an opposite-sex effect could be inferred for female respondents across measures. This is complicated by the fact that male respondents also showed lighter HAM preferences across all evaluative measures, implying variations in participant gender and/or sexual orientation were not central to evaluative responses. However, in Study 3, male respondents exhibited breaking times indicative of colorist preferences towards highly attractive female (HAF) targets, independent of the latter’s orientation. Assuming b-CFS was resilient to deliberative influence, we speculate that differences in breaking times might reflect optimized perceptual mechanisms geared toward identifying faces with high reproductive potential. HAM biases among females and HAF biases among males suggest that ‘skin tone’ was conditional to the presence of facial configurations with high reproductive potential. 

We inferred participants' sexual orientation from a 6-item scale, which showed that the majority of respondents were attracted to the opposite sex [p. 13, l. 14-23; p. 26, l. 3-7]. Responses to this survey are summarized on [Figure 3]. We could not directly examine the influence of sexual orientation on these effects given the qualitative and exploratory nature of survey responses, which have yet to be validated. We inferred a heterosexual orientation across males and females who respectively were attracted to ‘feminine-females’ and ‘masculine-males’, which constituted the majority of our sample. The single measure where male and female respondents significantly varied was b-CFS, which led to the speculation that breaking times indicative of colorist biases may be more pronounced for opposite-sex faces. Because our male participant pool in Study 3 was quite limited however, we caution against drawing population-level inferences [p. 31, l. 7-8]. 

===

Now, addressing some minor corrections (e.g., typos):

4. On page 15, lines 9 and 10, the authors refer to mean reaction times, but these should be referred to as response times. Additionally, they should include the standard deviations alongside the means. – Changed ‘reaction times’ to ‘response times’. The associated means are now included with standard deviations.

5. On page 16, line 10, the word "differed" has been omitted. – Added

6. On page 16, line 15, the phrase "to earlier" is unclear and requires further clarification. – Removed.

7. On page 17, line 12, the contraction "it's" should be replaced with the possessive form "its." – Removed

8. On page 20, line 4, when reporting the age range of the participants, the authors should also provide the mean age, standard deviation, and age range. – Included overall age range, alongside mean and SD ages for individual studies

9. On page 28, line 15, it says “who happened to sampled”. Please, correct. – Corrected to “who happened to be sampled”

10. On page 28, line 21, there is a point that should be a comma. – Corrected

In summary, this manuscript, consisting of three studies, addresses an intriguing question of social relevance. The research design is robust, the analyses are rigorous, and the discussion of the findings is persuasive and articulate. I am prepared to support the publication of this manuscript once the authors have addressed the concerns I have raised. – 

Thank you for your constructive comments. We hope the revisions have satisfactorily addressed your concerns.

References

Lanfranco, R. C., Rabagliati, H., & Carmel, D. (2023a). Assessing the influence of emotional expressions on perceptual sensitivity to faces overcoming interocular suppression. Emotion, 1–21. https://doi.org/10.1037/emo0001215

Lanfranco, R. C., Rabagliati, H., & Carmel, D. (2023b). The importance of awareness in face processing: A critical review of interocular suppression studies. Behavioural Brain Research, 437, 114116. https://doi.org/10.1016/j.bbr.2022.114116

Lanfranco, R. C., Stein, T., Rabagliati, H., & Carmel, D. (2022). Gaze direction and face orientation modulate perceptual sensitivity to faces under interocular suppression. Scientific Reports, 12(1), Article 1. https://doi.org/10.1038/s41598-022-11717-4

Pournaghdali, A., & Schwartz, B. L. (2020). Continuous flash suppression: Known and unknowns. Psychonomic Bulletin & Review, 27(6), 1071–1103. https://doi.org/10.3758/s13423-020-01771-2

Stein, T. (2019). The Breaking Continuous Flash Suppression Paradigm: Review, evaluation, and outlook. In G. Hesselmann (Ed.), Transitions between Consciousness and Unconsciousness (pp. 1–38). Routledge. https://www.taylorfrancis.com/chapters/breaking-continuous-flash-suppression-paradigm-timo-stein/10.4324/9780429469688-1

Stein, T., & Peelen, M. V. (2021). Dissociating conscious and unconscious influences on visual detection effects. Nature Human Behaviour, 5(5), 612–624. https://doi.org/10.1038/s41562-020-01004-5

Thank you for the references. These have been integrated into the revised manuscript.

Reviewer 3 comments:

Thank you for the opportunity to review the manuscript.

I believe it has considerable scientific importance with great potential for publication. However, the way the text is currently organized makes it difficult to understand the work. Thus, I provide below comments that I consider important to improve the work, assisting in its final publication. Mainly because it is three studies in one, the way this information is presented generates some confusion. 

Thank you for your commendation. We have substantially revised the manuscript in accordance with your suggestions. We hope the revisions are to your satisfaction. 

In the abstract, for example, I missed the methodological aspects and the conclusion of the study. I believe that the main work of the authors will be the construction of topics such as introduction, methodology, discussion and conclusion in a way that brings together the information of the work in a general way.

I believe that the keywords chosen do not reflect well the main objective of the study. I suggest that the authors could re-evaluate.

The revised abstract provides additional details about our methodology and findings. We have updated our keywords to accord more clearly with our study.

In the introduction, I missed previous studies that talk about the topic, demonstrating the importance of the study being conducted. Also, the introduction is confusing. It starts by talking about the topic, then addresses topics of the methodology used, then goes back to the objective of the study. My suggestion is: to make a general introduction about the study carried out, starting from the most general aspects to the most specific (citing previous studies, for example), ending with the objective and hypotheses of the study, being able to inform in a very brief way that to achieve these objectives three studies were carried out.

The Introduction has been revised to better align with the reviewer's suggestions. We start with a general overview of 'colorism', citing relevant studies before transitioning to the research objectives and current hypotheses [p. 4-8]. We point out that research into colorist preferences in psychological science has been comparatively sparse compared to research on racial attitudes, which the present research aimed to address [p. 4, l. 13-20]. Methodological details, which were initially interspersed in the Introduction, have been relocated to the appropriate sections under the Method [p. 13-17]. 

All methodological aspects are better described in the methodology section. Regarding the methodology, I believe that the authors can also carry out a general methodology with subtopics for the methodology used in each study carried out. I believe that the inclusion of information on these methodological aspects is very important, such as the inclusion and exclusion criteria of the sample, the period of data collection, how it was carried out online, what care the researchers used to avoid response bias in filling out these forms, among others.

The Methodology section has been restructured as per the reviewer’s suggestions. The general methodology section now contains subtopics detailing each study [pp. 8-18]. We have included information on the inclusion and exclusion criteria, specifying that all students enrolled in the Psychology program from September 2021 to July 2022 were eligible to participate. The period of data collection for each study is also stated, ranging from November 2021 to June 2022 [p. 8-9]. Issues with the online implementation using Google Forms is addressed under the Limitations on [p. 35, l. 3-14]. 

Better describe the analyses performed in each study, which variables were being evaluated, whether sample calculation was performed, whether the assumptions for performing the tests were met, etc.

On [pp. 9-10], we describe sensitivity analyses that informed our sample could detect moderate-to-large effects with 80% power at a 5% error rate. We mention all students registered in the Psychology program were eligible to take part. In the Introduction [pp. 5-8], we elaborate on the variables selected, and highlight the functional distinction between performances generated under both unconstrained and constrained conditions. We detail our planned tests near the end of the Introduction on [p.8 , l. 3 - 9], highlighting their robustness to parametric violations.

In my opinion, a general discussion uniting the characteristics of the three studies is more interesting than three discussions and a general discussion at the end. Mainly because in the specific discussion of the studies, the authors are only resuming the results and are not bringing the studies that corroborate or refute these results. 

The manuscript now includes a single, unified General Discussion that synthesizes the findings and theoretical considerations from all three studies, placing them in the context of existing literature [p. 30-39]. 

Another tip for the general discussion would be to resume the objective and hypotheses of the study, to speak briefly and without reporting the numerical data again, what the main results found, and to discuss these results based on the literature.

We commence the Discussion with a restatement of our study's objectives and hypotheses, providing a broad outline of our main findings and don’t restate numerical specifics. The results are related to existing literature [p. 34, l.5 – 22].

In the conclusion, I suggest that the authors can provide the conclusions of the study, without using references. It is the moment that the researcher can bring his perspectives on what was found and what remains to be researched, providing suggestions for future research.

Sincerely.

The revised Conclusion section is devoid of external references, focusing solely on summarizing the key findings of our study [p. 41]. We offer our perspectives on the implications of these findings and suggest avenues for future research.

---

## [Decision Letter · Decision Letter 1]

8 Dec 2023

Intra-group differences in skin tone influence evaluative and perceptual face processing

PONE-D-23-08925R1

Dear Dr. Amd,

We’re pleased to inform you that your manuscript has been judged scientifically suitable for publication and will be formally accepted for publication once it meets all outstanding technical requirements.

Kind regards,

Thiago P. Fernandes, PhD

Academic Editor

PLOS ONE

Additional Editor Comments (optional):

Thank you for your efforts in addressing the concerns; commended for your respect and dedication in this matter.

Reviewers' comments:

Reviewer's Responses to Questions

**Comments to the Author**

1. If the authors have adequately addressed your comments raised in a previous round of review and you feel that this manuscript is now acceptable for publication, you may indicate that here to bypass the “Comments to the Author” section, enter your conflict of interest statement in the “Confidential to Editor” section, and submit your "Accept" recommendation.

Reviewer #1: All comments have been addressed

Reviewer #2: All comments have been addressed

Reviewer #3: All comments have been addressed

2. Is the manuscript technically sound, and do the data support the conclusions?

Reviewer #1: Yes

Reviewer #2: Yes

Reviewer #3: Yes

3. Has the statistical analysis been performed appropriately and rigorously? 

Reviewer #1: Yes

Reviewer #2: Yes

Reviewer #3: Yes

4. Have the authors made all data underlying the findings in their manuscript fully available?

Reviewer #1: Yes

Reviewer #2: Yes

Reviewer #3: Yes

5. Is the manuscript presented in an intelligible fashion and written in standard English?

Reviewer #1: Yes

Reviewer #2: Yes

Reviewer #3: Yes

6. Review Comments to the Author

Reviewer #1: The authors have adequately addressed my comments. The quality of the manuscript has significantly improved, and I am pleased to recommend this work for publication.

Reviewer #2: The author has effectively addressed all of my comments. Upon thorough examination of their responses and revisions, I am pleased to endorse this manuscript for publication.

Reviewer #3: All my comments have been carefully reviewed and I believe the manuscript is ready for final publication.

7. PLOS authors have the option to publish the peer review history of their article (what does this mean?). If published, this will include your full peer review and any attached files.

Reviewer #1: No

Reviewer #2: **Yes: **Renzo Lanfranco

Reviewer #3: No

---

## [Editor Report · Acceptance letter]

20 Dec 2023

PONE-D-23-08925R1 

PLOS ONE

Dear Dr. Amd, 

I'm pleased to inform you that your manuscript has been deemed suitable for publication in PLOS ONE. Congratulations! Your manuscript is now being handed over to our production team.

Kind regards, 

on behalf of

Dr. Thiago P. Fernandes 

Academic Editor

PLOS ONE